# Mixed Linear Regression with Multiple Components

**Kai Zhong** [1]     **Prateek Jain** [2]     **Inderjit S. Dhillon** [3]
[1,3] University of Texas at Austin      [2] Microsoft Research India
[1] `zhongkai@ices.utexas.edu`,     [2] `prajain@microsoft.com`
[3] `inderjit@cs.utexas.edu`

## Abstract

In this paper, we study the mixed linear regression (MLR) problem, where the goal is to recover multiple underlying linear models from their unlabeled linear measurements. We propose a non-convex objective function which we show is *locally strongly convex* in the neighborhood of the ground truth. We use a tensor method for initialization so that the initial models are in the local strong convexity region. We then employ general convex optimization algorithms to minimize the objective function. To the best of our knowledge, our approach provides first exact recovery guarantees for the MLR problem with $K \geq 2$ components. Moreover, our method has near-optimal computational complexity $\widetilde{O}(Nd)$ as well as near-optimal sample complexity $\widetilde{O}(d)$ for *constant K*. Furthermore, we show that our non-convex formulation can be extended to solving the *subspace clustering* problem as well. In particular, when initialized within a small constant distance to the true subspaces, our method converges to the global optima (and recovers true subspaces) in time *linear* in the number of points. Furthermore, our empirical results indicate that even with random initialization, our approach converges to the global optima in linear time, providing speed-up of up to two orders of magnitude.

## 1   Introduction

The mixed linear regression (MLR) [7, 9, 29] models each observation as being generated from one of the $K$ unknown linear models; the identity of the generating model for each data point is also unknown. MLR is a popular technique for capturing non-linear measurements while still keeping the models simple and computationally efficient. Several widely-used variants of linear regression, such as piecewise linear regression [14, 28] and locally linear regression [8], can be viewed as special cases of MLR. MLR has also been applied in time-series analysis [6], trajectory clustering [15], health care analysis [11] and phase retrieval [4]. See [27] for more applications.

In general, MLR is NP-hard [29] with the hardness arising due to lack of information about the model labels (model from which a point is generated) as well as the model parameters. However, under certain statistical assumptions, several recent works have provided poly-time algorithms for solving MLR [2, 4, 9, 29]. But most of the existing recovery gurantees are restricted either to mixtures with $K = 2$ components [4, 9, 29] or require $poly(1/\epsilon)$ samples/time to achieve $\epsilon$-approximate solution [7, 24] (analysis of [29] for two components can obtain $\epsilon$ approximate solution in $\log(1/\epsilon)$ samples). Hence, solving the MLR problem with $K \geq 2$ mixtures while using near-optimal number of samples and computational time is still an open question.

In this paper, we resolve the above question under standard statistical assumptions for *constant* many mixture components $K$. To this end, we propose the following smooth objective function as a

surrogate to solve MLR:

$$f(\boldsymbol{w}_1, \boldsymbol{w}_2, \cdots, \boldsymbol{w}_K) := \sum_{i=1}^{n} \Pi_{k=1}^{K} (y_i - \boldsymbol{x}_i^T \boldsymbol{w}_k)^2, \tag{1}$$

where $\{(\boldsymbol{x}_i, y_i) \in \mathbb{R}^{d+1}\}_{i=1,2,\cdots,N}$ are the data points and $\{\boldsymbol{w}_k\}_{k=1,2,\cdots,K}$ are the model parameters. The intuition for this objective is that the objective value is zero when $\{\boldsymbol{w}_k\}_{k=1,2,\cdots,K}$ is the global optima and $y$'s do not contain any noise. Furthermore, the objective function is smooth and hence less prone to getting stuck in arbitrary saddle points or oscillating between two points. The standard EM [29] algorithm instead makes a "sharp" selection of mixture component and hence the algorithm is more likely to oscillate or get stuck. This intuition is reflected in Figure 1 (d) which shows that with random initialization, EM algorithm routinely gets stuck at poor solutions, while our proposed method based on the above objective still converges to the global optima.

Unfortunately, the above objective function is non-convex and is in general prone to poor saddle points, local minima. However under certain standard assumptions, we show that the objective is locally strongly convex (Theorem 1) in a small basin of attraction near the optimal solution. Moreover, the objective function is smooth. Hence, we can use gradient descent method to achieve linear rate of convergence to the global optima. But, we will need to initialize the optimization algorithm with an iterate which lies in a small ball around the optima. To this end, we modify the tensor method in [2, 7] to obtain a "good" initialization point. Typically, tensor methods require computation of third and higher order moments which leads to significantly worse sample complexity in terms of data dimensionality $d$. However, for the special case of MLR, we provide a small modification of the standard tensor method that achieves nearly optimal sample and time complexity bounds for constant $K$ (see Theorem 3). More concretely, our approach requires $\tilde{O}(d(K \log d)^K)$ many samples and requires $\tilde{O}(Nd)$ computational time; note the exponential dependence on $K$. Also for constant $K$, the method has nearly optimal sample and time complexity.

**Subspace clustering**: MLR can be viewed as a special case of subspace clustering (SC), since each regressor-response pair lies in the subspace determined by this pair's model parameters. However, solving MLR using SC approaches is intractable because the dimension of each subspace is only one less than the ambient dimension, which will easily violate the conditions for the recovery guarantees of most methods (see e.g. Table 1 in [23] for the conditions of different methods). Nonetheless, our objective for MLR easily extends to the subspace clustering problem. That is, given data points $\{\boldsymbol{z}_i \in \mathbb{R}^d\}_{i=1,2,\cdots,N}$, the goal is to minimize the following objective w.r.t. $K$ subspaces (each of dimension at most $r$):

$$\min_{U_k \in \mathbb{O}^{d \times r}, k=1,2,\cdots,K} f(U_1, U_2, \cdots, U_K) = \sum_{i=1}^{N} \Pi_{k=1}^{K} \langle I_d - U_k U_k^T, \boldsymbol{z}_i \boldsymbol{z}_i^T \rangle. \tag{2}$$

$U_k$ denotes the basis spanned by $k$-th estimated subspace and $\mathbb{O}^{d \times r} \subset \mathbb{R}^{d \times r}$ denotes the set of orthonormal matrices, i.e., $U^T U = I$ if $U \in \mathbb{O}^{d \times r}$. We propose a power-method style algorithm to alternately optimize (2) w.r.t $\{U_k\}_{k=1,2,\cdots,K}$, which takes only $O(rdN)$ time compared with $O(dN^2)$ for the state-of-the-art methods, e.g. [13, 22, 23].

Although EM with power method [4] shares the same computational complexity as ours, there is no convergence guarantee for EM to the best of our knowledge. In contrast, we provide local convergence guarantee for our method. That is, if $N = \tilde{O}(rK^K)$ and if data satisfies certain standard assumptions, then starting from an initial point $\{U_k\}_{k=1,\cdots,K}$ that lies in a small ball of constant radius around the globally optimal solution, our method converges super-linearly to the globally optimal solution. Unfortunately, our existing analyses do not provide global convergence guarantee and we leave it as a topic for future work. Interestingly, our empirical results indicated that even with randomly initialized $\{U_k\}_{k=1,\cdots,K}$, our method is able to recover the true subspace *exactly* using nearly $O(rK)$ samples.

We summarize our contributions below:

**(1) MLR**: We propose a non-convex continuous objective function for solving the mixed linear regression problem. To the best of our knowledge, our algorithm is the first work that can handle $K \geq 2$ components with global convergence guarantee in the noiseless case (Theorem 4). Our algorithm has near-optimal linear (in $d$) sample complexity and near-optimal computational complexity; however, our sample complexity dependence on $K$ is exponential.

**(2) Subspace Clustering**: We extend our objective function to subspace clustering, which can be optimized efficiently in $O(rdN)$ time compared with $O(dN^2)$ for state-of-the-art methods. We also provide a small basin of attraction in which our iterates converge to the global optima at super-linear rate (Theorem 5).

## 2 Related Work

**Mixed Linear Regression**:
*EM algorithm* without careful initialization is only guaranteed to have local convergence [4, 21, 29]. [29] proposed a grid search method for initialization. However, it is limited to the two-component case and seems non-trivial to extend to multiple components. It is known that exact minimization for each step of EM is not scalable due to the $O(d^2N + d^3)$ complexity. Alternatively, we can use EM with gradient update, whose local convergence is also guaranteed by [4] but only in the two-symmetric-component case, i.e., when $\boldsymbol{w}_2 = -\boldsymbol{w}_1$.

*Tensor Methods* for MLR were studied by [7, 24]. [24] approximated the third-order moment directly from samples with Gaussian distribution, while [7] learned the third-order moment from a low-rank linear regression problem. Tensor methods can obtain the model parameters to any precision $\epsilon$ but requires $1/\epsilon^2$ time/samples. Also, tensor methods can handle multiple components but suffer from high sample complexity and high computational complexity. For example, the sample complexity required by [7] and [24] is $O(d^6)$ and $O(d^3)$ respectively. On the other hand, the computational burden mainly comes from the operation on tensor, which costs at least $O(d^3)$ for a very simple tensor evaluation. [7] also suffers from the slow nuclear norm minimization when estimating the second and third order moments. In contrast, we use tensor method only for initialization, i.e., we require $\epsilon$ to be a certain constant. Moreover, with a simple trick, we can ensure that the sample and time complexity of our initialization step is only linear in $d$ and $N$.

*Convex Formulation.* Another approach to guarantee the recovery of the parameters is to relax the non-convex problem to convex problem. [9] proposed a convex formulation of MLR with two components. The authors provide upper bounds on the recovery errors in the noisy case and show their algorithm is information-theoretically optimal. However, the convex formulation needs to solve a nuclear norm function under linear constraints, which leads to high computational cost. The extension from two components to multiple components for this formulation is also not straightforward.

**Subspace Clustering**:
Subspace clustering [13, 17, 22, 23] is an important data clustering problem arising in many research areas. The most popular subspace clustering algorithms, such as [13, 17, 23], are based on a two-stage algorithm – first finding a neighborhood for each data point and then clustering the points given the neighborhood. The first stage usually takes at least $O(dN^2)$ time, which is prohibitive when $N$ is large. On the other hand, several methods such as K-subspaces clustering [18], K-SVD [1] and online subspace clustering [25] do have linear time complexity $O(rdN)$ per iteration, however, there are no global or local convergence guarantees. In contrast, we show locally superlinear convergence result for an algorithm with computational complexity $O(rdN)$. Our empirical results indicate that random initialization is also sufficient to get to the global optima; we leave further investigation of such an algorithm for future work.

## 3 Mixed Linear Regression with Multiple Components

In this paper, we assume the dataset $\{(\boldsymbol{x}_i, y_i) \in \mathbb{R}^{d+1}\}_{i=1,2,\cdots,N}$ is generated by,

$$z_i \sim \text{multinomial}(\boldsymbol{p}), \quad \boldsymbol{x}_i \sim \mathcal{N}(0, I_d), \quad y_i = \boldsymbol{x}_i^T \boldsymbol{w}_{z_i}^*, \tag{3}$$

where $\boldsymbol{p}$ is the proportion of different components satisfying $\boldsymbol{p}^T \boldsymbol{1} = 1$, $\{\boldsymbol{w}_k^* \in \mathbb{R}^d\}_{k=1,2,\cdots,K}$ are the ground truth parameters. The goal is to recover $\{\boldsymbol{w}_k^*\}_{k=1,2,\cdots,K}$ from the dataset. Our analysis is based on noiseless cases but we illustrate the empirical performance of our algorithm for the noisy cases, where $y_i = \boldsymbol{x}_i^T \boldsymbol{w}_{z_i}^* + e_i$ for some noise $e_i$ (see Figure 1).

**Notation.** We use $[N]$ to denote the set $\{1, 2, \cdots, N\}$ and $S_k \subset [N]$ to denote the index set of the samples that come from $k$-th component. Define $p_{\min} := \min_{k \in [K]}\{p_k\}$, $p_{\max} := \max_{k \in [K]}\{p_k\}$. Define $\Delta \boldsymbol{w}_j := \boldsymbol{w}_j - \boldsymbol{w}_j^*$ and $\Delta \boldsymbol{w}_{kj}^* := \boldsymbol{w}_k^* - \boldsymbol{w}_j^*$. Define $\Delta_{\min} := \min_{j \neq k}\{\|\Delta \boldsymbol{w}_{jk}^*\|\}$ and

$\Delta_{\max} := \max_{j \neq k}\{\|\Delta w_{jk}^*\|\}$. We assume $\frac{\Delta_{\min}}{\Delta_{\max}}$ is independent of the dimension $d$. Define $w := [w_1; w_2; \cdots; w_K] \in \mathbb{R}^{Kd}$. We denote $w^{(t)}$ as the parameters at $t$-th iteration and $w^{(0)}$ as the initial parameters. For simplicity, we assume there are $p_k N$ samples from the $k$-th model in any random subset of $N$ samples. We use $\mathbb{E}[\![X]\!]$ to denote the expectation of a random variable $X$. Let $T \in \mathbb{R}^{d \times d \times d}$ be a tensor and $T_{ijk}$ be the $i, j, k$-th entry of $T$. We say a tensor is supersymmetric if $T_{ijk}$ is invariant under any permutation of $i, j, k$. We also use the same notation $T$ to denote the multi-array map from three matrices, $A, B, C \in \mathbb{R}^{d \times r}$, to a new tensor: $[T(A, B, C)]_{i,j,k} = \sum_{p,q,l} T_{pql} A_{pi} B_{qj} C_{lk}$. We say a tensor $T$ is rank-one if $T = a \otimes b \otimes c$, where $T_{ijk} = a_i b_j c_k$. We use $\|A\|$ denote the spectral norm of the matrix $A$ and $\sigma_i(A)$ to denote the $i$-th largest singular value of $A$. For tensors, we use $\|T\|_{op}$ to denote the operator norm for a supersymmetric tensor $T$, $\|T\|_{op} := \max_{\|a\|=1} |T(a, a, a)|$. We use $T_{(1)} \in \mathbb{R}^{d \times d^2}$ to denote the matricizing of $T$ in the first order, i.e., $[T_{(1)}]_{i,(j-1)d+k} = T_{ijk}$. Throughout the paper, we use $\widetilde{O}(d)$ to denote $O(d \times \text{polylog}(d))$. We assume $K$ is a constant in general. However, if some numbers depend on $K^K$, we will explicitly present it in the big $O$ notation. For simplicity, we just include higher-order terms of $K$ and ignore lower-order terms, e.g., $O((2K)^{2K})$ may be replaced by $O(K^K)$.

## 3.1 Local Strong Convexity

In this section, we analyze the Hessian of objective (1).

**Theorem 1** (Local Positive Definiteness). *Let $\{x_i, y_i\}_{i=1,2,\cdots,N}$ be sampled from the MLR model (3). Let $\{w_k\}_{k=1,2,\cdots,K}$ be independent of the samples and lie in the neighborhood of the optimal solution, i.e.,*

$$\|\Delta w_k\| := \|w_k - w_k^*\| \leq c_m \Delta_{\min}, \forall k \in [K], \tag{4}$$

*where $c_m = O(p_{\min}(3K)^{-K}(\Delta_{\min}/\Delta_{\max})^{2K-2})$, $\Delta_{\min} = \min_{j \neq k}\{\|w_j^* - w_k^*\|\}$ and $\Delta_{\max} = \max_{j \neq k}\{\|w_j^* - w_k^*\|\}$. Let $P \geq 1$ be a constant. Then if $N \geq O((PK)^K d \log^{K+2}(d))$, w.p. $1 - O(Kd^{-P})$, we have,*

$$\frac{1}{8} p_{\min} N \Delta_{\min}^{2K-2} I \preceq \nabla^2 f(w + \delta w) \preceq 10 N (3K)^K \Delta_{\max}^{2K-2} I, \tag{5}$$

*for any $\delta w := [\delta w_1; \delta w_2; \cdots; \delta w_K]$ satisfying $\|\delta w_k\| \leq c_f \Delta_{\min}$, where $c_f = O(p_{\min}(3K)^{-K} d^{-K+1}(\Delta_{\min}/\Delta_{\max})^{2K-2})$.*

The above theorem shows the Hessians of a small neighborhood around a fixed $\{w_k\}_{k=1,2,\cdots,K}$, which is close enough to the optimum, are positive definiteness (PD). The conditions on $\{w_k\}_{k=1,\cdots,K}$ and $\{\delta w_k\}_{k=1,\cdots,K}$ are different. $\{w_k\}_{k=1,\cdots,K}$ are required to be independent of samples and in a ball of radius $c_m \Delta_{\min}$ centered at the optimal solution. On the other hand, $\{\delta w_k\}_{k=1,2,\cdots,K}$ can be dependent on the samples but are required to be in a smaller ball of radius $c_f \Delta_{\min}$. The conditions are natural as if $\Delta_{\min}$ is very small then distinguishing between $w_k^*$ and $w_{k'}^*$ is not possible and hence Hessians will not be PD w.r.t both the components.

To prove the theorem, we decompose the Hessian of Eq. (1) into multiple blocks, $(\nabla f)_{jl} = \frac{\partial^2 f}{\partial w_j \partial w_l} \in \mathbb{R}^{d \times d}$. When $w_k \to w_k^*$ for all $k \in [K]$, the diagonal blocks of the Hessian will be strictly positive definite. At the same time, the off-diagonal blocks will be close to zeros. The blocks are approximated by the samples using matrix Bernstein inequality. The detailed proof can be found in Appendix A.2.

Traditional analysis of optimization methods on strongly convex functions, such as gradient descent, requires the Hessians of all the parameters are PD. Theorem 1 implies that when $w_k = w_k^*$ for all $k = 1, 2, \cdots, K$, a small basin of attraction around the optimum is strongly convex as formally stated in the following corollary.

**Corollary 1** (Strong Convexity near the Optimum). *Let $\{x_i, y_i\}_{i=1,2,\cdots,N}$ be sampled from the MLR model (3). Let $\{w_k\}_{k=1,2,\cdots,K}$ lie in the neighborhood of the optimal solution, i.e.,*

$$\|w_k - w_k^*\| \leq c_f \Delta_{\min}, \forall k \in [K], \tag{6}$$

*where $c_f = O(p_{\min}(3K)^{-K} d^{-K+1}(\Delta_{\min}/\Delta_{\max})^{2K-2})$. Then, for any constant $P \geq 1$, if $N \geq O((PK)^K d \log^{K+2}(d))$, w.p. $1 - O(Kd^{-P})$, the objective function $f(w_1, w_2, \cdots, w_K)$ in Eq. (1) is strongly convex. In particular, w.p. $1 - O(Kd^{-P})$, for all $w$ satisfying Eq. (6),*

$$\frac{1}{8} p_{\min} N \Delta_{\min}^{2K-2} I \preceq \nabla^2 f(w) \preceq 10 N (3K)^K \Delta_{\max}^{2K-2} I. \tag{7}$$

The strong convexity of Corollary 1 only holds in the basin of attraction near the optimum that has diameter in the order of $O(d^{-K+1})$, which is too small to be achieved by our initialization method (in Sec. 3.2) using $\tilde{O}(d)$ samples. Next, we show by a simple construction, the linear convergence of gradient descent (GD) with resampling is still guaranteed when the solution is initialized in a much larger neighborhood.

**Theorem 2** (Convergence of Gradient Descent). *Let $\{\boldsymbol{x}_i, y_i\}_{i=1,2,\cdots,N}$ be sampled from the MLR model* (3). *Let $\{\boldsymbol{w}_k\}_{k=1,2,\cdots,K}$ be independent of the samples and lie in the neighborhood of the optimal solution, defined in Eq.* (4). *One iteration of gradient descent can be described as, $\boldsymbol{w}^+ = \boldsymbol{w} - \eta\nabla f(\boldsymbol{w})$, where $\eta = 1/(10N(3K)^K\Delta_{\max}^{2K-2})$. Then, if $N \geq O(K^K d \log^{K+2}(d))$, w.p. $1 - O(Kd^{-2})$,*

$$\|\boldsymbol{w}^+ - \boldsymbol{w}^*\|^2 \leq (1 - \frac{p_{\min}\Delta_{\min}^{2K-2}}{80(3K)^K\Delta_{\max}^{2K-2}})\|\boldsymbol{w} - \boldsymbol{w}^*\|^2 \tag{8}$$

**Remark.** The linear convergence Eq. (8) requires the resampling of the data points for each iteration. In Sec. 3.3, we combine Corollary 1, which doesn't require resampling when the iterate is sufficiently close to the optimum, to show that there exists an algorithm using a finite number of samples to achieve any solution precision.

To prove Theorem 2, we prove the PD properties on a line between a current iterate and the optimum by constructing a set of anchor points and then apply traditional analysis for the linear convergence of gradient descent. The detailed proof can be found in Appendix A.3.

### 3.2 Initialization via Tensor method

In this section, we propose a tensor method to initialize the parameters. We define the second-order moment $M_2 := \mathbb{E}[\![y^2(\boldsymbol{x} \otimes \boldsymbol{x} - I)]\!]$ and the third-order moments, $M_3 := \mathbb{E}[\![y^3\boldsymbol{x} \otimes \boldsymbol{x} \otimes \boldsymbol{x}]\!] - \sum_{j\in[d]}\mathbb{E}[\![y^3(\boldsymbol{e}_j \otimes \boldsymbol{x} \otimes \boldsymbol{e}_j + \boldsymbol{e}_j \otimes \boldsymbol{e}_j \otimes \boldsymbol{x} + \boldsymbol{x} \otimes \boldsymbol{e}_j \otimes \boldsymbol{e}_j)]\!]$. According to Lemma 6 in [24], $M_2 = \sum_{k=[K]} 2p_k\boldsymbol{w}_k^* \otimes \boldsymbol{w}_k^*$ and $M_3 = \sum_{k=[K]} 6p_k\boldsymbol{w}_k^* \otimes \boldsymbol{w}_k^* \otimes \boldsymbol{w}_k^*$. Therefore by calculating the eigendecomposition of the estimated moments, we are able to recover the parameters to any precision provided enough samples. Theorem 8 of [24] needs $O(d^3)$ sample complexity to obtain the model parameters with certain precision. Such high sample complexity comes from the tensor concentration bound. However, we find the problem of tensor eigendecomposition in MLR can be reduced to $\mathbb{R}^{K\times K\times K}$ space such that the sample complexity and computational complexity are $O(\text{poly}(K))$. Our method is similar to the whitening process in [7, 19]. However, [7] needs $O(d^6)$ sample complexity due to the nuclear-norm minimization problem, while ours requires only $\widetilde{O}(d)$. For this sample complexity, we need assume the following,

**Assumption 1.** *The following quantities, $\sigma_K(M_2)$, $\|M_2\|$, $\|M_3\|_{op}^{2/3}$, $\sum_{k\in[K]} p_k\|\boldsymbol{w}_k^*\|^2$ and $(\sum_{k\in[K]} p_k\|\boldsymbol{w}_k^*\|^3)^{2/3}$, have the same order of $d$, i.e., the ratios between any two of them are independent of $d$.*

The above assumption holds when $\{\boldsymbol{w}_k^*\}_{k=1,2,\cdots,K}$ are orthonormal to each other.

We formally present the tensor method in Algorithm 1 and its theoretical guarantee in Theorem 3.

**Theorem 3.** *Under Assumption 1, if $|\Omega| \geq O(d\log^2(d) + \log^4(d))$, then w.p. $1 - O(d^{-2})$, Algorithm 1 will output $\{\boldsymbol{w}_k^{(0)}\}_{k=1}^K$ that satisfies,*

$$\|\boldsymbol{w}_k^{(0)} - \boldsymbol{w}_k^*\| \leq c_m\Delta_{\min}, \forall k \in [K]$$

*which falls in the locally PD region, Eq.* (4), *in Theorem 1.*

The proof can be found in Appendix B.2. Forming $\hat{M}_2$ explicitly will cost $O(Nd^2)$ time, which is expensive when $d$ is large. We can compute each step of the power method without explicitly forming $\hat{M}_2$. In particular, we alternately compute $\hat{Y}^{(t+1)} = \sum_{i\in\Omega_{M_2}} y_i^2(\boldsymbol{x}_i(\boldsymbol{x}_i^TY^{(t)}) - Y^{(t)})$ and let $Y^{(t+1)} = \text{QR}(\hat{Y}^{(t+1)})$. Now each power method iteration only needs $O(KNd)$ time. Furthermore, the number of iterations needed will be a constant, since power method has linear convergence rate and we don't need very accurate solution. For the proof of this claim, we refer

---

**Algorithm 1** Initialization for MLR via Tensor Method

---

**Input:** $\{\boldsymbol{x}_i, y_i\}_{i \in \Omega}$

**Output:** $\{\boldsymbol{w}_k^{(0)}\}_{k=1}^K$

1: Partition the dataset $\Omega$ into $\Omega = \Omega_{M_2} \cup \Omega_2 \cup \Omega_3$ with $|\Omega_{M_2}| = O(d \log^2(d))$, $|\Omega_2| = O(d \log^2(d))$ and $|\Omega_3| = O(\log^4(d))$

2: Compute the approximate top-$K$ eigenvectors, $Y \in \mathbb{R}^{d \times K}$, of the second-order moment, $\hat{M}_2 := \frac{1}{|\Omega_{M_2}|} \sum_{i \in \Omega_{M_2}} y_i^2 (\boldsymbol{x}_i \otimes \boldsymbol{x}_i - I)$, by the power method.

3: Compute $\hat{R}_2 = \frac{1}{2|\Omega_2|} \sum_{i \in \Omega_2} y_i^2 (Y^T \boldsymbol{x}_i \otimes Y^T \boldsymbol{x}_i - I)$.

4: Compute the whitening matrix $\hat{W} \in \mathbb{R}^{K \times K}$ of $\hat{R}_2$, i.e., $\hat{W} = \hat{U}_2 \hat{\Lambda}_2^{-1/2} \hat{U}_2^T$, where $\hat{R}_2 = \hat{U}_2 \hat{\Lambda}_2 \hat{U}_2^T$ is the eigendecomposition of $\hat{R}_2$.

5: Compute $\hat{R}_3 = \frac{1}{6|\Omega_3|} \sum_{i \in \Omega_3} y_i^3 (\boldsymbol{r}_i \otimes \boldsymbol{r}_i \otimes \boldsymbol{r}_i - \sum_{j \in [K]} \boldsymbol{e}_j \otimes \boldsymbol{r}_i \otimes \boldsymbol{e}_j - \sum_{j \in [K]} \boldsymbol{e}_j \otimes \boldsymbol{e}_j \otimes \boldsymbol{r}_i - \sum_{j \in [K]} \boldsymbol{r}_i \otimes \boldsymbol{e}_j \otimes \boldsymbol{e}_j)$, where $\boldsymbol{r}_i = Y^T \boldsymbol{x}_i$ for all $i \in \Omega_3$.

6: Compute the eigenvalues $\{\hat{a}_k\}_{k=1}^K$ and the eigenvectors $\{\hat{\boldsymbol{v}}_k\}_{k=1}^K$ of the whitened tensor $\hat{R}_3(\hat{W}, \hat{W}, \hat{W}) \in \mathbb{R}^{K \times K \times K}$ by using the robust tensor power method [2].

7: Return the estimation of the models, $\boldsymbol{w}_k^{(0)} = Y(\hat{W}^T)^\dagger (\hat{a}_k \hat{\boldsymbol{v}}_k)$

---

to the proof of Lemma 10 in Appendix B. Next we compute $\hat{R}_2$ using $O(KNd)$ and compute $\hat{W}$ in $O(K^3)$ time. Computing $\hat{R}_3$ takes $O(KNd + K^3N)$ time. The robust tensor power method takes $O(\text{poly}(K)\text{polylog}(d))$ time. In summary, the computational complexity for the initialization is $O(KdN + K^3N + \text{poly}(K)\text{polylog}(d)) = \widetilde{O}(dN)$.

### 3.3 Global Convergence Algorithm

We are now ready to show the complete algorithm, Algorithm 2, that has global convergence guarantee. We use $f_\Omega(\boldsymbol{w})$ to denote the objective function Eq. (1) generated from a subset of the dataset $\Omega$, i.e., $f_\Omega(\boldsymbol{w}) = \sum_{i \in \Omega} \Pi_{k=1}^K (y_i - \boldsymbol{x}_i^T \boldsymbol{w}_k)^2$.

**Theorem 4** (Global Convergence Guarantee). *Let $\{\boldsymbol{x}_i, y_i\}_{i=1,2,\cdots,N}$ be sampled from the MLR model* (3) *with $N \geq O(d(K\log(d))^{2K+3})$. Let the step size $\eta$ be smaller than a positive constant. Then given any precision $\epsilon > 0$, after $T = O(\log(d/\epsilon))$ iterations, w.p. $1 - O(Kd^{-2}\log(d))$, the output of Algorithm 2 satisfies*

$$\|\boldsymbol{w}^{(T)} - \boldsymbol{w}^*\| \leq \epsilon \Delta_{\min}.$$

The detailed proof is in Appendix B.3. The computational complexity required by our algorithm is near-optimal: (a) tensor method (Algorithm 1) is carefully employed such that only $O(dN)$ computation is needed; (b) gradient descent with resampling is conducted in $\log(d)$ iterations to push the iterate to the next phase; (c) gradient descent without resampling is finally executed to achieve any precision with $\log(1/\epsilon)$ iterations. Therefore the total computational complexity is $O(dN \log(d/\epsilon))$. As shown in the theorem, our algorithm can achieve any precision $\epsilon > 0$ without any sample complexity dependency on $\epsilon$. This follows from Corollary 1 that shows local strong convexity of objective (1) with a fixed set of samples. By contrast, tensor method [7, 24] requires $O(1/\epsilon^2)$ samples and EM algorithm requires $O(\log(1/\epsilon))$ samples [4, 29].

## 4 Subspace Clustering (SC)

The mixed linear regression problem can be viewed as clustering $N$ $(d+1)$-dimensional data points, $\boldsymbol{z}_i = [\boldsymbol{x}_i, y_i]^T$, into one of the $K$ subspaces, $\{\boldsymbol{z} : [\boldsymbol{w}_k^*, -1]^T \boldsymbol{z} = 0\}$ for $k \in [K]$. Assume we have data points $\{\boldsymbol{z}_i\}_{i=1,2,\cdots,N}$ sampled from the following model,

$$a_i \sim \text{multinomial}(\boldsymbol{p}), \ \boldsymbol{s}_i \sim \mathcal{N}(0, I_r), \ \boldsymbol{z}_i = U_{a_i}^* \boldsymbol{s}_i, \tag{9}$$

where $\boldsymbol{p}$ is the proportion of samples from different subspaces and satisfies $\boldsymbol{p}^T \boldsymbol{1} = 1$ and $\{U_k^*\}_{k=1,2,\cdots,K}$ are the bases of the ground truth subspaces. We can solve Eq. (2) by alternately minimizing over $U_k$ when fixing the others, which is equivalent to finding the top-$r$ eigenvectors

of $\sum_{i=1}^{N} \alpha_i^k \boldsymbol{z}_i \boldsymbol{z}_i^T$, where $\alpha_i^k = \Pi_{j\neq k}\langle I_d - U_j U_j^T, \boldsymbol{z}_i \boldsymbol{z}_i^T\rangle$. When the dimension is high, it is very expensive to compute the exact top-$r$ eigenvectors. A more efficient way is to use one iteration of the power method (aka subspace iteration), which only takes $O(KdN)$ computational time per iteration. We present our algorithm in Algorithm 3.

We show Algorithm 3 will converge to the ground truth when the initial subspaces are sufficiently close to the underlying subspaces. Define $D(\hat{U}, \hat{V}) := \frac{1}{\sqrt{2}}\|UU^T - VV^T\|_F$ for some $\hat{U}, \hat{V} \in \mathbb{R}^{d\times r}$, where $U, V$ are orthogonal bases of $Span(\hat{U}), Span(\hat{V})$ respectively. Define $D_{\max} := \max_{j\neq q} D(U_q^*, U_j^*)$, $D_{\min} := \min_{j\neq q} D(U_q^*, U_j^*)$.

**Theorem 5.** *Let $\{\boldsymbol{z}_i\}_{i=1,2,\cdots,N}$ be sampled from subspace clustering model (9). If $N \geq O(r(K\log(r))^{2K+2})$ and the initial parameters $\{U_k^0\}_{k\in[K]}$ satisfy*

$$\max_k\{D(U_k^*, U_k^0)\} \leq c_s D_{\min}, \tag{10}$$

*where $c_s = O(p_{\min}/p_{\max}(3K)^{-K}(D_{\min}/D_{\max})^{2K-3})$, then w.p. $1 - O(Kr^{-2})$, the sequence $\{U_1^t, U_2^t, \cdots, U_K^t\}_{t=1,2,\cdots}$ generated by Algorithm 3 converges to the ground truth superlinearly. In particular, for $\Delta_t := \max_k\{D(U_k^*, U_k^t)\}$,*

$$\Delta_{t+1} \leq \Delta_t^2/(2c_s D_{\min}) \leq \frac{1}{2}\Delta_t.$$

We refer to Appendix C.2 for the proof. Compared to other methods, our sample complexity only depends on the dimension of each subspace linearly. We refer to Table 1 in [23] for a comparison of conditions for different methods. Note that if $D_{\min}/D_{\max}$ is independent of $r$ or $d$, then the initialization radius $c_s$ is a constant. However, initialization within the required distance to the optima is still an open question; tensor methods do not apply in this case. Interestingly, our experiments seem to suggest that our proposed method converges to the global optima (in the setting considered in the above theorem).

| **Algorithm 2** Gradient Descent for MLR | **Algorithm 3** Power Method for SC |
|---|---|
| **Input:** $\{\boldsymbol{x}_i, y_i\}_{i=1,2,\cdots,N}$, step size $\eta$. | **Input:** data points $\{\boldsymbol{z}_i\}_{i=1,2,\cdots,N}$ |
| **Output:** $\boldsymbol{w}$ | **Output:** $\{U_k\}_{k\in[K]}$ |
| 1: Partition the dataset into $\{\Omega^{(t)}\}_{t=0,1,\cdots,T_0+1}$ | 1: Some initialization, $\{U_k^0\}_{k\in[K]}$. |
| 2: Initialize $\boldsymbol{w}^{(0)}$ by Algorithm 1 with $\Omega^{(0)}$ | 2: Partition the data into $\{\Omega^{(t)}\}_{t=0,1,2,\cdots,T}$. |
| 3: **for** $t = 1, 2, \cdots, T_0$ **do** | 3: **for** $t = 0, 1, 2, \cdots, T$ **do** |
| 4: $\quad \boldsymbol{w}^{(t)} = \boldsymbol{w}^{(t-1)} - \eta\nabla f_{\Omega^{(t)}}(\boldsymbol{w}^{(t-1)})$ | 4: $\quad \alpha_i = \Pi_{j=1}^K \langle I_d - U_j^t U_j^{tT}, \boldsymbol{z}_i \boldsymbol{z}_i^T\rangle, i \in \Omega^{(t)}$ |
| 5: **for** $t = T_0+1, T_0+2, \cdots, T_0+T_1$ **do** | 5: $\quad$ **for** $k = 1, 2, \cdots, K$ **do** |
| 6: $\quad \boldsymbol{w}^{(t)} = \boldsymbol{w}^{(t-1)} - \eta\nabla f_{\Omega^{(T_0+1)}}(\boldsymbol{w}^{(t-1)})$ | 6: $\quad\quad \alpha_i^k = \alpha_i/\langle I_d - U_k^t U_k^{tT}, \boldsymbol{z}_i \boldsymbol{z}_i^T\rangle$ |
| | 7: $\quad\quad U_k^{t+1} \leftarrow QR(\sum_{i\in\Omega^{(t)}} \alpha_i^k \boldsymbol{z}_i \boldsymbol{z}_i^T U_k^t)$ |

# 5 Numerical Experiments

## 5.1 Mixed Linear Regression

In this section, we use synthetic data to show the properties of our algorithm that minimizes Eq. (1), which we call *LOSCO* (LOcally Strongly Convex Objective). We generate data points and parameters from standard normal distribution. We set $K = 3$ and $p_k = \frac{1}{3}$ for all $k \in [K]$. The error is defined as $\epsilon^{(t)} = \min_{\pi\in\text{Perm}([K])}\{\max_{k\in[K]}\|\boldsymbol{w}_{\pi(k)}^{(t)} - \boldsymbol{w}_k^*\|/\|\boldsymbol{w}_k^*\|\}$, where $\text{Perm}([K])$ is the set of all the permutation functions on the set $[K]$. The errors reported in the paper are averaged over 10 trials. In our experiments, we find there is no difference whether doing resampling or not. Hence, for simplicity, we use the original dataset for all the processes. We set both of two parameters in the robust tensor power method (denoted as $N$ and $L$ in Algorithm 1 in [2]) to be 100. The experiments are conducted in Matlab. After the initialization, we use alternating minimization (i.e., block coordinate descent) to exactly minimize the objective over $\boldsymbol{w}_k$ for $k = 1, 2, \cdots, K$ cyclicly.

Fig. 1(a) shows the recovery rate for different dimensions and different samples. We call the result of a trial is a successful recovery if $\epsilon^{(t)} < 10^{-6}$ for some $t < 100$. The recovery rate is the proportion of

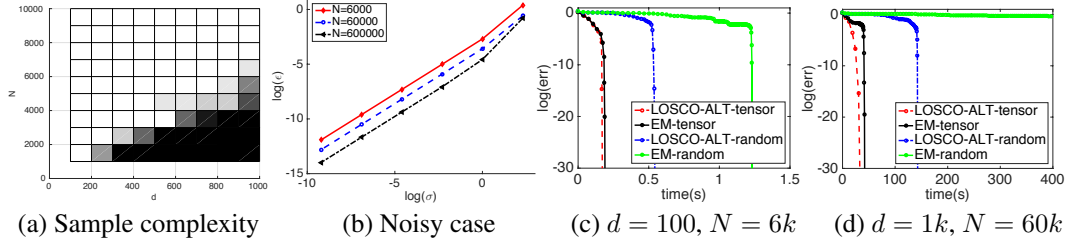

| (a) Sample complexity | (b) Noisy case | (c) $d = 100$, $N = 6k$ | (d) $d = 1k$, $N = 60k$ |

Figure 1: (a),(b): Empirical performance of our method. (c), (d): performance of our methods vs EM method. Our method with random initialization is signficantly better than EM with random initialization. Performance of the two methods is comparable when initialized with tensor method.

10 trials with successful recovery. As shown in the figure, the sample complexity for exact recovery is nearly linear to $d$. Fig. 1(b) shows the behavior of our algorithm in the noisy case. The noise is drawn from $e_i \in \mathcal{N}(0, \sigma^2)$, i.i.d., and $d$ is fixed as 100. As we can see from the figure, the solution error is almost proportional to the noise deviation. Comparing among different $N$'s, the solution error decreases when $N$ increases, so it seems consistent in presence of unbiased noise. We also illustrate the performance of our tensor initialization method in Fig. 2(a) in Appendix D.

We next compare with EM algorithm [29], where we alternately assign labels to points and exactly solve each model parameter according to the labels. EM has been shown to be very sensitive to the initialization [29]. The grid search initialization method proposed in [29] is not feasible here, because it only handles two components with a same magnitude. Therefore, we use random initialization and tensor initialization for EM. We compare our method with EM on convergence speed under different dimensions and different initialization methods. We use exact alternating minimization (LOSCO-ALT) to optimize our objective (1), which has similar computational complexity as EM. Fig. 1(c)(d) shows our method is competitive with EM on computational time, when it converges to the optima. In the case of (d), EM with random initialization doesn't converge to the optima, while our method still converges. In Appendix D, we will show some more experimental results.

Table 1: Time (sec.) comparison for different subspace clustering methods

| $N/K$ | SSC | SSC-OMP | LRR | TSC | NSN+spectral | NSN+GSR | **PSC** |
|---|---|---|---|---|---|---|---|
| 200 | 22.08 | 31.83 | 4.01 | 2.76 | 3.28 | 5.90 | 0.41 |
| 400 | 152.61 | 60.74 | 11.18 | 8.45 | 11.51 | 15.90 | 0.32 |
| 600 | 442.29 | 99.63 | 33.36 | 30.09 | 36.04 | 33.26 | 0.60 |
| 800 | 918.94 | 159.91 | 79.06 | 75.69 | 85.92 | 54.46 | 0.73 |
| 1000 | 1738.82 | 258.39 | 154.89 | 151.64 | 166.70 | 83.96 | 0.76 |

## 5.2 Subspace Clustering

In this section, we compare our subspace clustering method, which we call $PSC$ (Power method for Subspace Clustering), with state-of-the-art methods, SSC [13], SSC-OMP [12], LRR [22], TSC [17], NSN+spectral [23] and NSN+GSR [23] on computational time. We fix $K = 5$, $r = 30$ and $d = 50$. The ground truth $U_k^*$ is generated from Gaussian matrices. Each data point is a normalized Gaussian vector in their own subspace. Set $p_k = 1/K$. The initial subspace estimation is generated by orthonormlizing Gaussian matrices. The stopping criterion for our algorithm is that every point is clustered correctly, i.e., the clustering error (CE) (defined in [23]) is zero. We use publicly available codes for all the other methods (see [23] for the links).

As we shown from Table 1, our method is much faster than all other methods especially when $N$ is large. Almost all CE's corresponding to the results in Table 1 are very small, which are listed in Appendix D. We also illustrate CE's of our method for different $N$, $d$ and $r$ when fixing $K = 5$ in Fig. 6 of Appendix D, from which we see whatever the ambient dimension $d$ is, the clusters will be exactly recovered when $N$ is proportional to $r$.

**Acknowledgement:** This research was supported by NSF grants CCF-1320746, IIS-1546459 and CCF-1564000.

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
