[Supplementary Material]

# Appendix

## A Proofs of Local Convergence

We define a sequence of constants, $\{C_J\}_{J=0,1,\cdots}$, that satisfy

$$C_0 = 1, C_1 = 3, \text{ and } C_J = C_{J-1} + (4J^2 + 2J)C_{J-2} \text{ for } J \geq 2. \tag{11}$$

By construction, we can upper bound $C_J$,

$$
\begin{aligned}
C_J &\leq C_{J-2} + (4J^2 + 2J)C_{J-2} + (4(J-1)^2 + 2J - 2)C_{J-3} \\
&\leq C_{J-2} + (4J^2 + 2J)C_{J-2} + (4(J-1)^2 + 2J - 2)C_{J-2} \\
&\leq 8J^2 C_{J-2} \\
&\leq (3J)^J.
\end{aligned}
\tag{12}
$$

### A.1 Some Lemmata

We first introduce some lemmata, whose proofs can be found in Sec. A.4.

**Lemma 1** (Proposition 1.1 in [20]). *If $x \sim \mathcal{N}(0, I_d)$ and $\Sigma \in \mathbb{R}^{d \times d}$ is a fixed positive semi-definite matrix, then for all $t > 0$, w. p. $1 - e^{-t}$, we have*

$$x^T \Sigma x \leq \text{tr}(\Sigma) + 2\sqrt{\text{tr}(\Sigma^2)t} + 2\|\Sigma\|t.$$

By taking $t = P \log(d) + \log(n)$ for some $n \geq d$ and some constant $P \geq 1$, we have the following corollary.

**Corollary 2.** *If $x \sim \mathcal{N}(0, I_d)$ and $\Sigma \in \mathbb{R}^{d \times d}$ is a fixed positive semi-definite matrix, then for a fixed positive constant $P \geq 1$, we have, w. p. $1 - \frac{1}{n}d^{-P}$,*

$$x^T \Sigma x \leq \text{tr}(\Sigma) + 2\sqrt{\text{tr}(\Sigma^2)(P\log(d) + \log(n))} + 2\|\Sigma\|(P\log(d) + \log(n)) \leq (4P+5)\text{tr}(\Sigma)\log(n).$$

Setting $\Sigma = \beta\beta^T$ in Corollary 2, we have the following corollary.

**Corollary 3.** *If $x \sim \mathcal{N}(0, I_d)$ and $P \geq 1$ is a constant, then given any fixed $\beta \in \mathbb{R}^d$, w. p. $1 - \frac{1}{n}d^{-P}$, we have*

$$(\beta^T x)^2 \leq (4P+5)\|\beta\|^2 \log n.$$

Setting $\Sigma = I$ in Corollary 2, we have the following corollary.

**Corollary 4.** *If $x \sim \mathcal{N}(0, I_d)$ and $P \geq 1$ is a constant, then w. p. $1 - \frac{1}{n}d^{-P}$, we have*

$$\|x\|^2 \leq (4P+5)d\log n.$$

**Lemma 2** (Stein-type Lemma). *Let $x \sim \mathcal{N}(0, I_d)$ and $f(x)$ be a function of $x$ whose second derivative exists. Then*

$$\mathbb{E}\llbracket f(x)xx^T \rrbracket = \mathbb{E}\llbracket f(x) \rrbracket I + \mathbb{E}\llbracket \nabla^2 f(x) \rrbracket$$

**Lemma 3.** *Let $x \sim \mathcal{N}(0, I_d)$ and $A_k \succeq 0$ for all $k = 1, 2, \cdots, K$, then*

$$\Pi_{k=1}^K \text{tr}(A_k)I \preceq \mathbb{E}\llbracket \Pi_{k=1}^K (x^T A_k x)xx^T \rrbracket \preceq C_K \Pi_{k=1}^K \text{tr}(A_k)I, \tag{13}$$

*where $C_K$ is a constant depending only on $K$, which is defined in Eq. (11).*

**Lemma 4.** *Let $x_i \sim \mathcal{N}(0, I_d)$ i.i.d., for all $i \in [n]$ and $A_k \succeq 0$ for all $k = 1, 2, \cdots, K$. Let $B := \mathbb{E}\llbracket \Pi_{k=1}^K (x^T A_k x)xx^T \rrbracket$, $B_i := \Pi_{k=1}^K (x_i^T A_k x_i)x_i x_i^T$ and $\hat{B} = \frac{1}{n}\sum_{i=1}^n B_i$.*

*If $n \geq O(\frac{1}{\delta^2}\log^K(\frac{1}{\delta})(PK)^K d \log^{K+1} d)$ and $\delta > \frac{\sqrt{4KC_{2K+1}}}{\sqrt{nd^P}}$ for some $0 < \delta \leq 1$ and $P \geq 1$, then w.p. $1 - O(Kd^{-P})$, we have*

$$\|\hat{B} - B\| \leq \delta\|B\|. \tag{14}$$

**Lemma 5.** *Let $x \sim \mathcal{N}(0, I_d)$. Then given $\boldsymbol{\beta}, \boldsymbol{\gamma} \in \mathbb{R}^d$ and $A_k \succeq 0$ for all $k = 1, 2, \cdots, K$, we have*

$$\|\boldsymbol{\beta}\|\|\boldsymbol{\gamma}\|\Pi_{k=1}^K \operatorname{tr}(A_k) \leq \|\mathbb{E}[(\boldsymbol{\beta}^T \boldsymbol{x})(\boldsymbol{\gamma}^T \boldsymbol{x})\Pi_{k=1}^K(\boldsymbol{x}^T A_k \boldsymbol{x})\boldsymbol{x}\boldsymbol{x}^T]\| \leq \sqrt{3C_{2K+1}}\|\boldsymbol{\beta}\|\|\boldsymbol{\gamma}\|\Pi_{k=1}^K \operatorname{tr}(A_k) I. \tag{15}$$

**Lemma 6.** *Let $\boldsymbol{x}_i \sim \mathcal{N}(0, I_d)$ i.i.d., for all $i \in [n]$, $\boldsymbol{\beta}, \boldsymbol{\gamma} \in \mathbb{R}^d$ and $A_k \succeq 0$ for all $k = 1, 2, \cdots, K$. Let $B := \mathbb{E}[(\boldsymbol{\beta}^T \boldsymbol{x})(\boldsymbol{\gamma}^T \boldsymbol{x})\Pi_{k=1}^K(\boldsymbol{x}^T A_k \boldsymbol{x})\boldsymbol{x}\boldsymbol{x}^T]$, $B_i := (\boldsymbol{\beta}^T \boldsymbol{x}_i)(\boldsymbol{\gamma}^T \boldsymbol{x}_i)\Pi_{k=1}^K(\boldsymbol{x}_i^T A_k \boldsymbol{x}_i)\boldsymbol{x}_i \boldsymbol{x}_i^T$ and $\hat{B} = \frac{1}{n}\sum_{i=1}^n B_i$.*

*If $n \geq O(\frac{1}{\delta^2}\log^{K+1}(1/\delta)(PK)^K d\log^{K+2}(d))$, $\delta > \frac{\sqrt{8KC_{2K+3}}}{\sqrt{nd^P}}$ for some $0 < \delta \leq 1$ and $P \geq 1$, then w.p. $1 - O(Kd^{-P})$, we have*

$$\|\hat{B} - B\| \leq \delta\|B\|. \tag{16}$$

**Lemma 7.** *If $n \geq c\log^{K+1}(c)K^{4K}d\log^{K+2}(d)$, where $c$ is a constant, then $n \geq cd\log d\log^{K+1}(n)$.*

## A.2  Proof of Theorem 1

*Proof.* Denote the Hessian of Eq. (1), $H \in \mathbb{R}^{Kd \times Kd}$. Let $H = \sum_i H_i$, where

$$H_i := \begin{bmatrix} H_i^{11} & H_i^{12} & \cdots & H_i^{1K} \\ H_i^{21} & H_i^{22} & \cdots & H_i^{2K} \\ & & \ddots & \\ H_i^{K1} & H_i^{K2} & \cdots & H_i^{KK} \end{bmatrix} \tag{17}$$

For diagonal blocks,

$$H_i^{jj} := 2\big(\Pi_{k\neq j}(y_i - (\boldsymbol{w}_k + \delta\boldsymbol{w}_k)^T \boldsymbol{x}_i)^2\big)\boldsymbol{x}_i \boldsymbol{x}_i^T \tag{18}$$

For off-diagonal blocks,

$$H_i^{jl} := 4(y_i - (\boldsymbol{w}_j + \delta\boldsymbol{w}_j)^T \boldsymbol{x}_i)(y_i - (\boldsymbol{w}_l + \delta\boldsymbol{w}_l)^T \boldsymbol{x}_i)\big(\Pi_{k\neq j, k\neq l}(y_i - (\boldsymbol{w}_k + \delta\boldsymbol{w}_k)^T \boldsymbol{x}_i)^2\big)\boldsymbol{x}_i \boldsymbol{x}_i^T \tag{19}$$

In the following we will show that when $\boldsymbol{w}_k$ is close to the optimal solution $\boldsymbol{w}_k^*$ and $\delta\boldsymbol{w}_k$ is small enough for all $k$, then $H$ will be positive definite w.h.p..

The main idea is to upper bound the off-diagonal blocks and lower bound the diagonal blocks because,

$$\begin{aligned}
\sigma_{min}(H) &= \min_{\sum_{j=1}^K \|\boldsymbol{a}_j\|^2 = 1} \sum_{j=1}^K \boldsymbol{a}_j^T H^{jj} \boldsymbol{a}_j + \sum_{j\neq l} 2\boldsymbol{a}_j^T H^{jl} \boldsymbol{a}_l \\
&\geq \min_{\sum_{j=1}^K \|\boldsymbol{a}_j\|^2 = 1} \sum_{j=1}^K \sigma_{min}(H^{jj})\|\boldsymbol{a}_j\|^2 - \sum_{j\neq l} \|H^{jl}\|\|\boldsymbol{a}_j\|\|\boldsymbol{a}_l\| \\
&\geq \min_j \{\sigma_{min}(H^{jj})\} - \max_{j\neq l}\{\|H^{jl}\|\}(K-1)(\sum_j \|\boldsymbol{a}_j\|) \\
&\geq \min_j \{\sigma_{min}(H^{jj})\} - (K-1)\max_{j\neq l}\{\|H^{jl}\|\}.
\end{aligned} \tag{20}$$

First consider the diagonal blocks. The idea is to decompose the diagonal blocks into two parts. The first one only contains $\boldsymbol{w}$ and doesn't contain $\delta\boldsymbol{w}$, so for this fixed $\boldsymbol{w}$ we apply Lemma 4 to bound this term. The second one depends on $\delta\boldsymbol{w}$. We find an upper bound for this term which only depends on the magnitude of $\delta\boldsymbol{w}$. Therefore, the bound will hold for any qualified $\delta\boldsymbol{w}$. Let's first define

$\{k_1, k_2, \cdots, k_{K-1}\} = [K]\backslash\{j\}.$

$$H^{jj} \succeq \sum_{i \in S_j} H_i^{jj}$$

$$= \sum_{i \in S_j} 2\big(\Pi_{s=1}^{K-1}(y_i - \boldsymbol{w}_{k_s}^T \boldsymbol{x}_i - \delta \boldsymbol{w}_{k_s}^T \boldsymbol{x}_i)^2\big)\boldsymbol{x}_i \boldsymbol{x}_i^T$$

$$\succeq \sum_{i \in S_j} 2\big((y_i - \boldsymbol{w}_{k_1}^T \boldsymbol{x}_i)^2 - 2|y_i - \boldsymbol{w}_{k_1}^T \boldsymbol{x}_i|\|\delta \boldsymbol{w}_{k_1}\|\|\boldsymbol{x}_i\|\big)\big(\Pi_{s=2}^{K-1}(y_i - \boldsymbol{w}_{k_s}^T \boldsymbol{x}_i - \delta \boldsymbol{w}_{k_s}^T \boldsymbol{x}_i)^2\big)\boldsymbol{x}_i \boldsymbol{x}_i^T$$

$$\succeq \underbrace{\sum_{i \in S_j} 2(y_i - \boldsymbol{w}_{k_1}^T \boldsymbol{x}_i)^2\big(\Pi_{s=2}^{K-1}(y_i - \boldsymbol{w}_{k_s}^T \boldsymbol{x}_i - \delta \boldsymbol{w}_{k_s}^T \boldsymbol{x}_i)^2\big)\boldsymbol{x}_i \boldsymbol{x}_i^T}_{F_1}$$

$$\underbrace{- \sum_{i \in S_j} 4\|\Delta \boldsymbol{w}_{jk_1}^* - \Delta \boldsymbol{w}_{k_1}\|\|\delta \boldsymbol{w}_{k_1}\|\|\boldsymbol{x}_i\|^2\big(\Pi_{s=2}^{K-1}(y_i - \boldsymbol{w}_{k_s}^T \boldsymbol{x}_i - \delta \boldsymbol{w}_{k_s}^T \boldsymbol{x}_i)^2\big)\boldsymbol{x}_i \boldsymbol{x}_i^T}_{E_1}$$

$$(21)$$

$$F_1 \succeq \sum_{i \in S_j} 2(y_i - \boldsymbol{w}_{k_1}^T \boldsymbol{x}_i)^2(y_i - \boldsymbol{w}_{k_2}^T \boldsymbol{x}_i)^2\big(\Pi_{s=3}^{K-1}(y_i - \boldsymbol{w}_{k_s}^T \boldsymbol{x}_i - \delta \boldsymbol{w}_{k_s}^T \boldsymbol{x}_i)^2\big)\boldsymbol{x}_i \boldsymbol{x}_i^T$$

$$- \sum_{i \in S_j} 4(y_i - \boldsymbol{w}_{k_1}^T \boldsymbol{x}_i)^2\|\Delta \boldsymbol{w}_{jk_2}^* - \Delta \boldsymbol{w}_{k_2}\|\|\delta \boldsymbol{w}_{k_2}\|\|\boldsymbol{x}_i\|^2\big(\Pi_{s=3}^{K-1}(y_i - \boldsymbol{w}_{k_s}^T \boldsymbol{x}_i - \delta \boldsymbol{w}_{k_s}^T \boldsymbol{x}_i)^2\big)\boldsymbol{x}_i \boldsymbol{x}_i^T$$

$$\succeq \underbrace{\sum_{i \in S_j} 2(y_i - \boldsymbol{w}_{k_1}^T \boldsymbol{x}_i)^2(y_i - \boldsymbol{w}_{k_2}^T \boldsymbol{x}_i)^2\big(\Pi_{s=3}^{K-1}(y_i - \boldsymbol{w}_{k_s}^T \boldsymbol{x}_i - \delta \boldsymbol{w}_{k_s}^T \boldsymbol{x}_i)^2\big)\boldsymbol{x}_i \boldsymbol{x}_i^T}_{F_2}$$

$$\underbrace{- \sum_{i \in S_j} 4\|\Delta \boldsymbol{w}_{jk_1}^* - \Delta \boldsymbol{w}_{k_1}\|^2\|\Delta \boldsymbol{w}_{jk_2}^* - \Delta \boldsymbol{w}_{k_2}\|\|\delta \boldsymbol{w}_{k_2}\|\|\boldsymbol{x}_i\|^4\big(\Pi_{s=3}^{K-1}(y_i - \boldsymbol{w}_{k_s}^T \boldsymbol{x}_i - \delta \boldsymbol{w}_{k_s}^T \boldsymbol{x}_i)^2\big)\boldsymbol{x}_i \boldsymbol{x}_i^T}_{E_2}$$

$$(22)$$

Similarly, we decompose $F_n = F_{n+1} - E_{n+1}$, for $n = 1, 2, \cdots, K - 1$. Then, recursively, we have

$$H^{jj} \succeq F_1 - E_1 \succeq F_2 - E_2 - E_1 \succeq \cdots \succeq F_{K-1} - E_{K-1} - E_{K-2} - \cdots - E_1 \qquad (23)$$

So $H^{jj}$ is decomposed into $F_{K-1}$, which contains only $\boldsymbol{w}$, and $E_1, E_2, \cdots, E_{K-1}$, each of which contains a separate term of $\|\delta \boldsymbol{w}\|$.

By Lemma 3 and Lemma 4,

$$E_1 \preceq 4 \sum_{i \in S_j} \|\Delta \boldsymbol{w}_{jk_1}^* - \Delta \boldsymbol{w}_{k_1}\|\|\delta \boldsymbol{w}_{k_1}\|(\Pi_{s=2}^{K-1}\|\Delta \boldsymbol{w}_{jk_s}^* - \Delta \boldsymbol{w}_{k_s} - \delta \boldsymbol{w}_{k_s}\|^2)\|\boldsymbol{x}_i\|^{2(K-1)}\boldsymbol{x}_i \boldsymbol{x}_i^T$$

$$\preceq 4c_f(1 + c_m + c_f)^{2K-3}\Pi_{k:k\neq j}\|\Delta \boldsymbol{w}_{jk}^*\|^2 \sum_{i \in S_j} \|\boldsymbol{x}_i\|^{2(K-1)}\boldsymbol{x}_i \boldsymbol{x}_i^T$$

$$\preceq 6c_f(1 + c_m + c_f)^{2K-3}\Pi_{k:k\neq j}\|\Delta \boldsymbol{w}_{jk}^*\|^2 p_j N C_{K-1} d^{K-1} I$$

$$(24)$$

and similarly, for all $r = 1, 2, \cdots, K - 1$,

$$E_r \preceq 6c_f(1 + c_m + c_f)^{2K-3}\Pi_{k:k\neq j}\|\Delta \boldsymbol{w}_{jk}^*\|^2 p_j N C_{K-1} d^{K-1} I. \qquad (25)$$

For $F_{K-1}$, we have

$$F_{K-1} = \sum_{i \in S_j} 2\big(\Pi_{k\neq j}(y_i - \boldsymbol{w}_k^T \boldsymbol{x}_i)^2\big)\boldsymbol{x}_i \boldsymbol{x}_i^T$$

$$\overset{\xi_1}{\succeq} p_j N \Pi_{k\neq j}\|\Delta \boldsymbol{w}_{jk}^* - \Delta \boldsymbol{w}_k\|^2 I$$

$$\succeq p_j N \Pi_{k\neq j}(\|\Delta \boldsymbol{w}_{jk}^*\| - \|\Delta \boldsymbol{w}_k\|)^2 I$$

$$\succeq p_j N (1 - c_m)^{2(K-1)}\Pi_{k\neq j}\|\Delta \boldsymbol{w}_{jk}^*\|^2 I$$

$$(26)$$

where $\xi_1$ is because of Lemma 3 and Lemma 4 by setting $A_k = (\Delta \boldsymbol{w}_{jk}^* - \Delta \boldsymbol{w}_k)(\Delta \boldsymbol{w}_{jk}^* - \Delta \boldsymbol{w}_k)^T$ and $\delta = 1/(2C_{K-1})$.

Now combining Eq. (26), Eq. (23) and Eq. (25), we can lower bound the eigenvalues of $H^{jj}$,

$$H^{jj} \succeq \left( (1 - c_m)^{2(K-1)} - 6c_f(K-1)(1 + c_m + c_f)^{2K-3}C_{K-1}d^{K-1} \right) p_j N \Pi_{k \neq j} \|\Delta \boldsymbol{w}_{jk}^*\|^2 I$$

(27)

Next consider the off-diagonal blocks for $j \neq l$,

$$\sum_{i \in S_q} H_i^{jl}$$

$$= \sum_{i \in S_q} 4(y_i - (\boldsymbol{w}_j + \delta \boldsymbol{w}_j)^T \boldsymbol{x}_i)(y_i - (\boldsymbol{w}_l + \delta \boldsymbol{w}_l)^T \boldsymbol{x}_i)\left( \Pi_{k \neq j, k \neq l}(y_i - (\boldsymbol{w}_k + \delta \boldsymbol{w}_k)^T \boldsymbol{x}_i)^2 \right) \boldsymbol{x}_i \boldsymbol{x}_i^T$$

$$\preceq \sum_{i \in S_q} 4(y_i - \boldsymbol{w}_j^T \boldsymbol{x}_i)(y_i - (\boldsymbol{w}_l + \delta \boldsymbol{w}_l)^T \boldsymbol{x}_i)\left( \Pi_{k \neq j, k \neq l}(y_i - (\boldsymbol{w}_k + \delta \boldsymbol{w}_k)^T \boldsymbol{x}_i)^2 \right) \boldsymbol{x}_i \boldsymbol{x}_i^T$$

$$+ \sum_{i \in S_q} 4\|\delta \boldsymbol{w}_j^T \boldsymbol{x}_i\| |y_i - (\boldsymbol{w}_l + \delta \boldsymbol{w}_l)^T \boldsymbol{x}_i| \left( \Pi_{k \neq j, k \neq l}(y_i - (\boldsymbol{w}_k + \delta \boldsymbol{w}_k)^T \boldsymbol{x}_i)^2 \right) \boldsymbol{x}_i \boldsymbol{x}_i^T$$

$$\preceq \sum_{i \in S_q} 4(y_i - \boldsymbol{w}_j^T \boldsymbol{x}_i)(y_i - \boldsymbol{w}_l^T \boldsymbol{x}_i)\left( \Pi_{k \neq j, k \neq l}(y_i - (\boldsymbol{w}_k + \delta \boldsymbol{w}_k)^T \boldsymbol{x}_i)^2 \right) \boldsymbol{x}_i \boldsymbol{x}_i^T$$

$$+ \sum_{i \in S_q} 4|y_i - \boldsymbol{w}_j^T \boldsymbol{x}_i| \|\delta \boldsymbol{w}_l^T \boldsymbol{x}_i\| \left( \Pi_{k \neq j, k \neq l}(y_i - (\boldsymbol{w}_k + \delta \boldsymbol{w}_k)^T \boldsymbol{x}_i)^2 \right) \boldsymbol{x}_i \boldsymbol{x}_i^T$$

$$+ \sum_{i \in S_q} 4\|\delta \boldsymbol{w}_j\| \|\boldsymbol{w}_q^* - \boldsymbol{w}_l - \delta \boldsymbol{w}_l)\| \left( \Pi_{k \neq j, k \neq l}\|\boldsymbol{w}_q^* - \boldsymbol{w}_k + \delta \boldsymbol{w}_k\|^2 \right) \|\boldsymbol{x}_i\|^{2(K-1)} \boldsymbol{x}_i \boldsymbol{x}_i^T$$

$$\preceq$$

$$\vdots$$

$$\preceq \sum_{i \in S_q} 4(y_i - \boldsymbol{w}_j^T \boldsymbol{x}_i)(y_i - \boldsymbol{w}_l^T \boldsymbol{x}_i)\left( \Pi_{k \neq j, k \neq l}(y_i - \boldsymbol{w}_k^T \boldsymbol{x}_i)^2 \right) \boldsymbol{x}_i \boldsymbol{x}_i^T$$

$$+ 8(K-1)c_f(1 + c_m + c_f)^{2K-3} \Delta_{max}^{2K-2} \sum_{i \in S_q} \|\boldsymbol{x}_i\|^{2(K-1)} \boldsymbol{x}_i \boldsymbol{x}_i^T$$

$$\preceq \sum_{i \in S_q} 4(y_i - \boldsymbol{w}_j^T \boldsymbol{x}_i)(y_i - \boldsymbol{w}_l^T \boldsymbol{x}_i)\left( \Pi_{k \neq j, k \neq l}(y_i - \boldsymbol{w}_k^T \boldsymbol{x}_i)^2 \right) \boldsymbol{x}_i \boldsymbol{x}_i^T$$

$$+ 12(K-1)c_f(1 + c_m + c_f)^{2K-3} \Delta_{max}^{2K-2} p_q N C_{K-1} d^{K-1} I$$

(28)

For the first term above,

$$\| \sum_{i \in S_q} 4(\boldsymbol{w}_q^* - \boldsymbol{w}_j)^T \boldsymbol{x}_i (\boldsymbol{w}_q^* - \boldsymbol{w}_l)^T \boldsymbol{x}_i \left( \Pi_{k \neq j, k \neq l}((\boldsymbol{w}_q^* - \boldsymbol{w}_k)^T \boldsymbol{x}_i)^2 \right) \boldsymbol{x}_i \boldsymbol{x}_i^T \|$$

$$\overset{\xi_1}{\leq} 6p_q N \| \mathbb{E}\left[\!\left[ (\boldsymbol{w}_q^* - \boldsymbol{w}_j)^T \boldsymbol{x}_i (\boldsymbol{w}_q^* - \boldsymbol{w}_l)^T \boldsymbol{x}_i \left( \Pi_{k \neq j, k \neq l}((\boldsymbol{w}_q^* - \boldsymbol{w}_k)^T \boldsymbol{x}_i)^2 \right) \boldsymbol{x}_i \boldsymbol{x}_i^T \right]\!\right] \|$$

(29)

$$\overset{\xi_2}{\leq} 6p_q N \sqrt{3C_{2K-3}} \|\boldsymbol{w}_q^* - \boldsymbol{w}_j\| \|\boldsymbol{w}_q^* - \boldsymbol{w}_l\| \left( \Pi_{k \neq j, k \neq l}\|\boldsymbol{w}_q^* - \boldsymbol{w}_k\|^2 \right)$$

$$\leq 6p_q N \sqrt{3C_{2K-3}} \|\Delta \boldsymbol{w}_{qj}^* - \Delta \boldsymbol{w}_j\| \|\Delta \boldsymbol{w}_{ql}^* - \Delta \boldsymbol{w}_l\| \left( \Pi_{k \neq j, k \neq l}\|\Delta \boldsymbol{w}_{qk}^* - \Delta \boldsymbol{w}_k\|^2 \right),$$

where $\xi_1$ is because of Lemma 6 and $\xi_2$ is because of Lemma 5.

We consider three cases: $\{q \neq j, q \neq l\}$, $q = j$ and $q = l$. When $q \neq j$ and $q \neq l$,

$$\|\Delta \boldsymbol{w}_{qj}^* - \Delta \boldsymbol{w}_j\| \|\Delta \boldsymbol{w}_{ql}^* - \Delta \boldsymbol{w}_l\| \left( \Pi_{k \neq j, k \neq l}\|\Delta \boldsymbol{w}_{qk}^* - \Delta \boldsymbol{w}_k\|^2 \right)$$

$$\leq (1 + c_m)^{2K-2} c_m^2 \|\Delta \boldsymbol{w}_{qj}^*\| \|\Delta \boldsymbol{w}_{ql}^*\| \left( \Pi_{k \neq j, k \neq l}\|\Delta \boldsymbol{w}_{qk}^*\|^2 \right)$$

(30)

When $q = j$,

$$\|\Delta \boldsymbol{w}_{qj}^* - \Delta \boldsymbol{w}_j\|\|\Delta \boldsymbol{w}_{ql}^* - \Delta \boldsymbol{w}_l\|\big(\Pi_{k \neq j, k \neq l}\|\Delta \boldsymbol{w}_{qk}^* - \Delta \boldsymbol{w}_k\|^2\big)$$
$$\leq (1 + c_m)^{2K-1} c_m \|\Delta \boldsymbol{w}_{qj}^*\|\|\Delta \boldsymbol{w}_{ql}^*\|\big(\Pi_{k \neq j, k \neq l}\|\Delta \boldsymbol{w}_{qk}^*\|^2\big) \tag{31}$$

For $q = l$, we have similar results. Therefore,

$$\begin{aligned}
\|H^{jl}\| &\leq \sum_{q=1}^{K} \|\sum_{i \in S_q} H_i^{jl}\| \\
&\leq \sum_q (1 + c_m)^{2K-1} c_m 6 p_q N \sqrt{3 C_{2K-3}} \Delta_{max}^{2K-2} \\
&\quad + \sum_q 12(K-1) c_f (1 + c_m + c_f)^{2K-3} p_q N C_{K-1} d^{K-1} \Delta_{max}^{2K-2} \\
&\leq (1 + c_m)^{2K-1} c_m 6 N \sqrt{3 C_{2K-3}} \Delta_{max}^{2K-2} \\
&\quad + 12(K-1) c_f (1 + c_m + c_f)^{2K-3} N C_{K-1} d^{K-1} \Delta_{max}^{2K-2}
\end{aligned} \tag{32}$$

Now we obtain the lower bound for the minimal eigenvalue of the Hessian. When $c_m \leq \frac{p_{min} \Delta_{min}^{2K-2}}{500 K \sqrt{C_{2K-3}} \Delta_{max}^{2K-2}}$ and $c_f \leq \frac{p_{min} \Delta_{min}^{2K-2}}{1000(K-1)^2 C_{K-1} d^{K-1} \Delta_{max}^{2K-2}}$, we have $(1 - c_m)^{2K-2} \geq (1 - \frac{1}{2K})^{2K-2} \geq \frac{1}{4}$, $(1 + c_m + c_f)^{2K-2} \leq 3$. Hence,

$$\|H^{jl}\| \leq \frac{1}{16(K-1)} p_{min} N \Delta_{min}^{2K-2}, \tag{33}$$

Combining Eq.(20), Eq.(27) and Eq.(33), we have

$$\sigma_{min}(H) \geq \frac{1}{8} p_{min} N \Delta_{min}^{2K-2}, \tag{34}$$

which is a positive constant.

In the following we upper bound the maximal eigenvalue of the Hessian.

$$\begin{aligned}
\sigma_{max}(H) &= \max_{\sum_{j=1}^{K} \|\boldsymbol{a}_j\|^2 = 1} \sum_{j=1}^{K} \boldsymbol{a}_j^T H^{jj} \boldsymbol{a}_j + \sum_{j \neq l} 2 \boldsymbol{a}_j^T H^{jl} \boldsymbol{a}_l \\
&\leq \max_{\sum_{j=1}^{K} \|\boldsymbol{a}_j\|^2 = 1} \sum_{j=1}^{K} \|(H^{jj})\|\|\boldsymbol{a}_j\|^2 + \sum_{j \neq l} \|H^{jl}\|\|\boldsymbol{a}_j\|\|\boldsymbol{a}_l\| \\
&\leq \max_j \{\|H^{jj}\|\} + \max_{j \neq l} \{\|H^{jl}\|\}(K-1)(\sum_j \|\boldsymbol{a}_j\|) \\
&\leq \max_j \{\|H^{jj}\|\} + (K-1) \max_{j \neq l} \{\|H^{jl}\|\}.
\end{aligned} \tag{35}$$

Consider the diagonal blocks and define $\{k_1, k_2, \cdots, k_{K-1}\} = [K] \backslash \{j\}$.

$$\begin{aligned}
H_i^{jj} &= 2\big(\Pi_{s=1}^{K-1}(y_i - \boldsymbol{w}_{k_s}^T \boldsymbol{x}_i - \delta \boldsymbol{w}_{k_s}^T \boldsymbol{x}_i)^2\big) \boldsymbol{x}_i \boldsymbol{x}_i^T \\
&\preceq 2\big((y_i - \boldsymbol{w}_{k_1}^T \boldsymbol{x}_i)^2 + 2|y_i - \boldsymbol{w}_{k_1}^T \boldsymbol{x}_i||\delta \boldsymbol{w}_{k_1}^T \boldsymbol{x}_i| + (\delta \boldsymbol{w}_{k_1}^T \boldsymbol{x}_i)^2\big) \big(\Pi_{s=2}^{K-1}(y_i - \boldsymbol{w}_{k_s}^T \boldsymbol{x}_i - \delta \boldsymbol{w}_{k_s}^T \boldsymbol{x}_i)^2\big) \boldsymbol{x}_i \boldsymbol{x}_i^T \\
&\preceq \underbrace{2(y_i - \boldsymbol{w}_{k_1}^T \boldsymbol{x}_i)^2 \big(\Pi_{s=2}^{K-1}(y_i - \boldsymbol{w}_{k_s}^T \boldsymbol{x}_i - \delta \boldsymbol{w}_{k_s}^T \boldsymbol{x}_i)^2\big) \boldsymbol{x}_i \boldsymbol{x}_i^T}_{\tilde{F}_1} \\
&\quad + \underbrace{2(2\|\Delta \boldsymbol{w}_{jk_1}^* - \Delta \boldsymbol{w}_{k_1}\| + \|\delta \boldsymbol{w}_{k_1}\|)\|\delta \boldsymbol{w}_{k_1}\|\|\boldsymbol{x}_i\|^2 \big(\Pi_{s=2}^{K-1}(y_i - \boldsymbol{w}_{k_s}^T \boldsymbol{x}_i - \delta \boldsymbol{w}_{k_s}^T \boldsymbol{x}_i)^2\big) \boldsymbol{x}_i \boldsymbol{x}_i^T}_{\tilde{E}_1}
\end{aligned} \tag{36}$$

For $\tilde{E}_1$,

$$\tilde{E}_1 \preceq 4 c_f (1 + c_m + c_f)^{2K-3} \Delta_{max}^{2K-2} \|\boldsymbol{x}_i\|^{2K-2} \boldsymbol{x}_i \boldsymbol{x}_i^T \tag{37}$$

For $\tilde{F}_1$,

$$\tilde{F}_1$$

$$\preceq \underbrace{2(y_i - \boldsymbol{w}_{k_1}^T \boldsymbol{x}_i)^2 (y_i - \boldsymbol{w}_{k_2}^T \boldsymbol{x}_i)^2 \left( \Pi_{s=3}^{K-1} (y_i - \boldsymbol{w}_{k_s}^T \boldsymbol{x}_i - \delta \boldsymbol{w}_{k_s}^T \boldsymbol{x}_i)^2 \right) \boldsymbol{x}_i \boldsymbol{x}_i^T}_{\tilde{F}_2}$$

$$+ \underbrace{2(y_i - \boldsymbol{w}_{k_1}^T \boldsymbol{x}_i)^2 |(2\Delta \boldsymbol{w}_{jk_2}^* - 2\Delta \boldsymbol{w}_{k_2} - \delta \boldsymbol{w}_{k_2})^T \boldsymbol{x}_i| |\delta \boldsymbol{w}_{k_2}^T \boldsymbol{x}_i| \left( \Pi_{s=3}^{K-1} (y_i - \boldsymbol{w}_{k_s}^T \boldsymbol{x}_i - \delta \boldsymbol{w}_{k_s}^T \boldsymbol{x}_i)^2 \right) \boldsymbol{x}_i \boldsymbol{x}_i^T}_{\tilde{E}_2}$$

$$(38)$$

We also have for $\tilde{E}_2$

$$\tilde{E}_2 \preceq 4c_f(1 + c_m + c_f)^{2K-3} \Delta_{max}^{2K-2} \|\boldsymbol{x}_i\|^{2K-2} \boldsymbol{x}_i \boldsymbol{x}_i^T \qquad (39)$$

Therefore, recursively, we have

$$H_i^{jj} \preceq \underbrace{2\Pi_{s=1}^{K-1} (y_i - \boldsymbol{w}_{k_s}^T \boldsymbol{x}_i)^2 \boldsymbol{x}_i \boldsymbol{x}_i^T}_{\tilde{F}_{K-1}}$$

$$+ 4Kc_f(1 + c_m + c_f)^{2K-3} \Delta_{max}^{2K-2} \|\boldsymbol{x}_i\|^{2K-2} \boldsymbol{x}_i \boldsymbol{x}_i^T \qquad (40)$$

Now applying Lemma 3 and Lemma 4,

$$H^{jj} = \sum_q \sum_{i \in S_q} H_i^{jj}$$

$$\preceq 6c_f K(1 + c_m + c_f)^{2K-3} N C_{K-1} d^{K-1} \Delta_{max}^{2K-2} I + \sum_q \sum_{i \in S_q} 2\left( \Pi_{k \neq q}((\Delta \boldsymbol{w}_{jk}^* - \Delta \boldsymbol{w}_k)^T \boldsymbol{x}_i)^2 \right) \boldsymbol{x}_i \boldsymbol{x}_i^T$$

$$\preceq 6c_f K(1 + c_m + c_f)^{2K-3} N C_{K-1} d^{K-1} \Delta_{max}^{2K-2} I + 3\sum_q p_q N C_{K-1} \left( \Pi_{k \neq q} \|\Delta \boldsymbol{w}_{jk}^* - \Delta \boldsymbol{w}_k\|^2 \right)$$

$$= 6c_f K(1 + c_m + c_f)^{2K-3} N C_{K-1} d^{K-1} \Delta_{max}^{2K-2} I$$

$$+ 3p_j N C_{K-1} (1 + c_m)^{2K-2} \left( \Pi_{k \neq j} \|\Delta \boldsymbol{w}_{jk}^*\|^2 \right)$$

$$+ 3\sum_{q:q \neq j} p_q N C_{K-1} c_m^2 (1 + c_m)^{2K-4} \left( \Pi_{k:k \neq j} \|\Delta \boldsymbol{w}_{jk}^*\|^2 \right)$$

$$\preceq 9N C_{K-1} \Delta_{max}^{2K-2} I$$

$$(41)$$

Combining the off-diagonal blocks bound in Eq. (33), applying union bound on the probabilities of the lemmata and Eq. (12) complete the proof. $\qquad \square$

## A.3 Proof of Theorem 2

We first introduce a corollary of Theorem 1, which shows the strong convexity on a line between a current iterate and the optimum.

**Corollary 5** (Positive Definiteness on the Line between $\boldsymbol{w}$ and $\boldsymbol{w}^*$). *Let* $\{\boldsymbol{x}_i, y_i\}_{i=1,2,\cdots,N}$ *be sampled from the MLR model* (3). *Let* $\{\boldsymbol{w}_k\}_{k=1,2,\cdots,K}$ *be independent of the samples and lie in the neighborhood of the optimal solution, defined in Eq.* (4). *Then, if* $N \geq O(K^K d \log^{K+2}(d))$, *w.p.* $1 - O(Kd^{-2})$, *for all* $\lambda \in [0, 1]$,

$$\frac{1}{8} p_{\min} N \Delta_{\min}^{2K-2} I \preceq \nabla^2 f(\lambda \boldsymbol{w}^* + (1 - \lambda)\boldsymbol{w}) \preceq 10N(3K)^K \Delta_{\max}^{2K-2} I. \qquad (42)$$

*Proof.* We set $d^{K-1}$ anchor points equally along the line $\lambda \boldsymbol{w}^* + (1 - \lambda)\boldsymbol{w}$ for $\lambda \in [0, 1]$. Then based on these anchors, according to Theorem 1, by setting $P = K + 1$, we complete the proof. $\qquad \square$

Now we show the proof of Theorem 2.

*Proof.* Let $\alpha := \frac{1}{8}p_{\min}N\Delta_{\min}^{2K-2}$ and $\beta := 10N(3K)^K\Delta_{\max}^{2K-2}$.

$$
\begin{aligned}
\|\boldsymbol{w}^+ - \boldsymbol{w}^*\|^2 &= \|\boldsymbol{w} - \eta\nabla f(\boldsymbol{w}) - \boldsymbol{w}^*\|^2 \\
&= \|\boldsymbol{w} - \boldsymbol{w}^*\|^2 - 2\eta\nabla f(\boldsymbol{w})^T(\boldsymbol{w} - \boldsymbol{w}^*) + \eta^2\|\nabla f(\boldsymbol{w})\|^2
\end{aligned}
\tag{43}
$$

$$
\begin{aligned}
\nabla f(\boldsymbol{w}) &= \left(\int_0^1 \nabla^2 f(\boldsymbol{w}^* + \gamma(\boldsymbol{w} - \boldsymbol{w}^*))d\gamma\right)(\boldsymbol{w} - \boldsymbol{w}^*) \\
&=: \hat{H}(\boldsymbol{w} - \boldsymbol{w}^*)
\end{aligned}
\tag{44}
$$

According to Corollary 5,

$$
\alpha I \preceq \hat{H} \preceq \beta I.
\tag{45}
$$

$$
\|\nabla f(\boldsymbol{w})\|^2 = (\boldsymbol{w} - \boldsymbol{w}^*)^T\hat{H}^2(\boldsymbol{w} - \boldsymbol{w}^*) \leq \beta(\boldsymbol{w} - \boldsymbol{w}^*)^T\hat{H}(\boldsymbol{w} - \boldsymbol{w}^*)
\tag{46}
$$

Therefore,

$$
\begin{aligned}
\|\boldsymbol{w}^+ - \boldsymbol{w}^*\|^2 &\leq \|\boldsymbol{w} - \boldsymbol{w}^*\|^2 - (-\eta^2\beta + 2\eta)(\boldsymbol{w} - \boldsymbol{w}^*)^T\hat{H}(\boldsymbol{w} - \boldsymbol{w}^*) \\
&\leq \|\boldsymbol{w} - \boldsymbol{w}^*\|^2 - (-\eta^2\beta + 2\eta)\alpha\|\boldsymbol{w} - \boldsymbol{w}^*\|^2 \\
&= \|\boldsymbol{w} - \boldsymbol{w}^*\|^2 - \frac{\alpha}{\beta}\|\boldsymbol{w} - \boldsymbol{w}^*\|^2 \\
&\leq (1 - \frac{\alpha}{\beta})\|\boldsymbol{w} - \boldsymbol{w}^*\|^2
\end{aligned}
\tag{47}
$$

where the third equality holds by setting $\eta = \frac{1}{\beta}$. $\qquad\square$

### A.4 Proof of the lemmata

#### A.4.1 Proof of Lemma 2

*Proof.* Let $g(\boldsymbol{x}) = \frac{1}{(2\pi)^{d/2}}e^{-\|\boldsymbol{x}\|^2/2}$ and we have $\boldsymbol{x}g(\boldsymbol{x})\mathrm{d}\boldsymbol{x} = -\mathrm{d}g(\boldsymbol{x})$.

$$
\begin{aligned}
\mathbb{E}\llbracket f(\boldsymbol{x})\boldsymbol{x}\boldsymbol{x}^T\rrbracket &= \int f(\boldsymbol{x})\boldsymbol{x}\boldsymbol{x}^Tg(\boldsymbol{x})\mathrm{d}\boldsymbol{x} \\
&= -\int f(\boldsymbol{x})(\mathrm{d}g(\boldsymbol{x}))\boldsymbol{x}^T \\
&= \int \nabla f(\boldsymbol{x})\boldsymbol{x}^Tg(\boldsymbol{x}))\mathrm{d}\boldsymbol{x} + \int f(\boldsymbol{x})g(\boldsymbol{x})I\mathrm{d}\boldsymbol{x} \\
&= -\int \nabla f(\boldsymbol{x})(\mathrm{d}g(\boldsymbol{x}))^T + \mathbb{E}\llbracket f(\boldsymbol{x})\rrbracket I \\
&= \mathbb{E}\llbracket \nabla^2 f(\boldsymbol{x})\rrbracket + \mathbb{E}\llbracket f(\boldsymbol{x})\rrbracket I
\end{aligned}
\tag{48}
$$

$\qquad\square$

#### A.4.2 Proof of Lemma 3

*Proof.* Let $G_K := \mathbb{E}\llbracket \Pi_{k=1}^K(\boldsymbol{x}^T A_k\boldsymbol{x})\boldsymbol{x}\boldsymbol{x}^T\rrbracket$. First we show the lower bound.

$$
\begin{aligned}
\sigma_{min}(G_K) &= \min_{\|\boldsymbol{a}\|=1}\mathbb{E}\llbracket \Pi_{k=1}^K(\boldsymbol{x}^T A_k\boldsymbol{x})(\boldsymbol{x}^T\boldsymbol{a})^2\rrbracket \\
&\geq \Pi_{k=1}^K\mathbb{E}\llbracket(\boldsymbol{x}^T A_k\boldsymbol{x})\rrbracket \min_{\|\boldsymbol{a}\|=1}\mathbb{E}\llbracket(\boldsymbol{x}^T\boldsymbol{a})^2\rrbracket \\
&= \Pi_{k=1}^K\mathrm{tr}(A_k)
\end{aligned}
\tag{49}
$$

Next, we show the upper bound. As we know, when $K = 1$, $G_1 = \mathrm{tr}(A_1)I + 2A_1$ and for any $K > 1$, $G_K$ should have an explicit closed-form. However, it is too complicated to derive and formulate it for general $K$. Fortunately we only need the property of Eq. (13) in our proofs. We

prove it by induction. First, it is obvious that Eq. (13) holds for $K = 1$ and $C_1 = 3$. We assume that, for any $J < K$, there exists a constant $C_J$ depending only on $J$, such that

$$G_J \preceq C_J \Pi_{k=1}^J \operatorname{tr}(A_k) I \tag{50}$$

Then by Stein-type lemma, Lemma 2,

$$
\begin{aligned}
G_K =& \mathbb{E}\big[\![\Pi_{k=1}^K (\boldsymbol{x}^T A_k \boldsymbol{x}) \boldsymbol{x} \boldsymbol{x}^T]\!\big] \\[4pt]
=& \mathbb{E}\big[\![\Pi_{k=1}^K (\boldsymbol{x}^T A_k \boldsymbol{x})]\!\big] I + 2 \sum_{j=1}^K \mathbb{E}\big[\![(\Pi_{k\neq j}^K (\boldsymbol{x}^T A_k \boldsymbol{x})) A_j]\!\big] \\[4pt]
&+ 4 \sum_{j,l:j\neq l} A_j \mathbb{E}\big[\![(\Pi_{k:k\neq j,k\neq l}(\boldsymbol{x}^T A_k \boldsymbol{x})) \boldsymbol{x} \boldsymbol{x}^T]\!\big] A_l \\[4pt]
\preceq& C_{K-1} \Pi_{k=1}^K \operatorname{tr}(A_k) I + 2 \sum_{j=1}^K C_{K-2} (\Pi_{k\neq j} \operatorname{tr}(A_k)) A_j \\[4pt]
&+ 4 \sum_{j,l:j\neq l} C_{K-2} \|A_j\|\|A_l\|\Pi_{k:k\neq j,k\neq l} \operatorname{tr}(A_k) I \\[4pt]
\preceq& \big(C_{K-1} + (2K + 4K^2)C_{K-2}\big)\Pi_{k=1}^K \operatorname{tr}(A_k) I
\end{aligned}
\tag{51}
$$

So $C_K = C_{K-1} + (4K^2 + 2K)C_{K-2}$. Note that $C_0 = 1$. $\qquad\square$

### A.4.3 Proof of Lemma 4

*Proof. Proof Sketch*: We use matrix Bernstein inequality to prove this lemma. However, the spectral norm of the random matrix $B_i$ is not uniformly bounded, which is required by matrix Bernstein inequality. So we define a new random matrix,

$$M_i := \mathbf{1}(\mathcal{E}_i)\Pi_{k=1}^K (\boldsymbol{x}_i^T A_k \boldsymbol{x}_i)\boldsymbol{x}_i \boldsymbol{x}_i^T,$$

where $\mathcal{E}_i$ is an event when $\|B_i\|$ is bounded, which will hold with high probability and $\mathbf{1}()$ is the indicate function of value 1 and 0, i.e., $\mathbf{1}(\mathcal{E}) = 1$ if $\mathcal{E}$ holds and $\mathbf{1}(\mathcal{E}) = 0$ otherwise. Then

$$\|\hat{B} - B\| \leq \|\hat{B} - \hat{M}\| + \|\hat{M} - M\| + \|M - B\|,$$

where $M = \mathbb{E}[\![M_i]\!]$ and $\hat{M} = \frac{1}{n} \sum_{i=1}^n M_i$. We show that

1. $\hat{M} = \hat{B}$ w.h.p. by the union bound
2. $\|\hat{M} - M\|$ is bounded by matrix Bernstein inequality
3. $\|M - B\|$ is bounded because $\mathbb{E}[\![\mathbf{1}(\mathcal{E}^c)]\!]$ is small.

*Proof Details:*

**Step 1.** First we show that $\|B_i\|$ is bounded w.h.p.. First,

$$\|B_i\| = \Pi_{k=1}^K (\boldsymbol{x}_i^T A_k \boldsymbol{x}_i)\|\boldsymbol{x}_i\|^2$$

Since $\boldsymbol{x} \sim \mathcal{N}(0, I_d)$, by Corollary 4, we have $\mathbb{P}[\![\|\boldsymbol{x}\|^2 \geq (4P + 5)d \log n]\!] \leq n^{-1}d^{-P}$. By Corollary 2, $\mathbb{P}[\![\boldsymbol{x}^T A_k \boldsymbol{x} > (4P + 5) \operatorname{tr}(A_k) \log n]\!] \leq n^{-1}d^{-P}$. Therefore w.p. $1 - (K + 1)n^{-1}d^{-P}$,

$$\|B_i\| \leq (4P + 5)^{K+1} \times (\Pi_{k=1}^K \operatorname{tr}(A_k))d \log^{K+1}(n).$$

Define

$$m := (4P + 5)^{K+1}(\Pi_{k=1}^K \operatorname{tr}(A_k))d \log^{K+1}(n). \tag{52}$$

and the event

$$\mathcal{E}_i = \{\|B_i\| \leq m\},$$

Let $\mathcal{E}^c$ be the complementary set of $\mathcal{E}$, thus $\mathbb{P}[\![\mathcal{E}_i^c]\!] \leq (K + 1)n^{-1}d^{-P}$. By union bound, w.p. $1 - (K + 1)d^{-P}$, $\|B_i\| \leq m$ for all $i \in [n]$ and $\hat{M} = \hat{B}$.

**Step 2.** Now we bound $\|\hat{M} - M\|$ by Matrix Bernstein's inequality[26].

Set $Z_i := M_i - M$. Thus $\mathbb{E}[\![Z_i]\!] = 0$ and $\|Z_i\| \leq 2m$. And

$$\|\mathbb{E}[\![Z_i^2]\!]\| = \|\mathbb{E}[\![M_i^2]\!] - M^2\| \leq \|\mathbb{E}[\![M_i^2]\!]\| + \|M^2\|$$

Since $M$ is PSD, $\|\mathbb{E}[\![M_i^2]\!]\| \leq m\|M\|$. Now by matrix Bernstein's inequality, for any $\delta > 0$,

$$\mathbb{P}\left[\!\!\left[\frac{1}{n}\|\sum_{i=1}^n Z_i\| \geq \delta\|M\|\right]\!\!\right] \leq 2d\exp(-\frac{\delta^2 n^2\|M\|^2/2}{mn\|M\| + 2mn\delta\|M\|/3}) = 2d\exp(-\frac{\delta^2 n\|M\|/2}{m + 2m\delta/3}) \tag{53}$$

Setting

$$n \geq (P+1)(\frac{4}{3\delta} + \frac{2}{\delta^2})m\|M\|^{-1}\log d, \tag{54}$$

we have w.p. at least $1 - 2d^{-P}$,

$$\|\frac{1}{n}\sum M_i - M\| \leq \delta\|M\| \tag{55}$$

**Step 3.** Now we bound $\|M - B\|$. For simplicity, we replace $\boldsymbol{x}_i$ by $\boldsymbol{x}$ and $\mathcal{E}_i$ by $\mathcal{E}$.

$$
\begin{aligned}
&\|M - B\| \\
=&\|\mathbb{E}[\![B_i\mathbf{1}(\mathcal{E}_i^c)]\!]\| \\
=& \max_{\|\boldsymbol{a}\|=1} \mathbb{E}[\![(\boldsymbol{a}^T\boldsymbol{x})^2\Pi_{k=1}^K(\boldsymbol{x}^T A_k\boldsymbol{x})\mathbf{1}(\mathcal{E}^c)]\!] \\
\overset{\zeta_1}{\leq}& \max_{\|\boldsymbol{a}\|=1} \mathbb{E}[\![(\boldsymbol{a}^T\boldsymbol{x})^4\Pi_{k=1}^K(\boldsymbol{x}^T A_k\boldsymbol{x})^2]\!]^{1/2}\mathbb{E}[\![\mathbf{1}(\mathcal{E}^c)]\!]^{1/2} \\
=& \max_{\|\boldsymbol{a}\|=1} \langle \boldsymbol{a}\boldsymbol{a}^T, \mathbb{E}[\![(\boldsymbol{x}^T\boldsymbol{a}\boldsymbol{a}^T\boldsymbol{x})\Pi_{k=1}^K(\boldsymbol{x}^T A_k\boldsymbol{x})^2\boldsymbol{x}\boldsymbol{x}^T]\!]\rangle^{1/2}\mathbb{E}[\![\mathbf{1}(\mathcal{E}^c)]\!]^{1/2} \\
\overset{\zeta_2}{\leq}& \max_{\|\boldsymbol{a}\|=1} \langle \boldsymbol{a}\boldsymbol{a}^T, C_{2K+1}\Pi_{k=1}^K \operatorname{tr}(A_k)^2 I\rangle^{1/2}\mathbb{E}[\![\mathbf{1}(\mathcal{E}^c)]\!]^{1/2} \\
\overset{\zeta_3}{\leq}& \frac{\sqrt{(K+1)C_{2K+1}}}{\sqrt{nd^P}}\Pi_{k=1}^K \operatorname{tr}(A_k)
\end{aligned} \tag{56}
$$

where $\zeta_1$ is from Holder's inequality, $\zeta_2$ is because of Lemma 3 and $\zeta_3$ is because $\mathbb{E}[\![\mathbf{1}(\mathcal{E}^c)]\!] = \mathbb{P}[\![\mathcal{E}^c]\!]$. Assume $n \geq 4(K+1)C_{2K+1}/d^P$, we have $\|M - B\| \leq \frac{1}{2}\|B\|$ and $\frac{3}{2}\|B\| \geq \|M\| \geq \frac{1}{2}\|B\|$. So combining this result with Eq. (52), Eq. (54), and Eq. (55), if

$$n \geq \max\{4(K+1)C_{2K+1}/d^P, c_1\frac{1}{\delta^2}(4P+5)^{K+2}d\log^{K+1}(n)\log d\}, \tag{57}$$

we obtain

$$\|\frac{1}{n}\sum M_i - M\| \leq \frac{1}{3}\delta\|M\| \leq \frac{1}{2}\delta\|B\|. \tag{58}$$

According to Lemma 7, $n \geq O(\frac{1}{\delta^2}\log^{K+1}(\frac{1}{\delta})(PK)^K d\log^{K+2} d)$ will imply Eq. (57). By further setting $\delta > \frac{\sqrt{4(K+1)C_{2K+1}}}{\sqrt{nd^P}}$, we have $\|M - B\| \leq \frac{1}{2}\delta\|B\|$, completing the proof.

$\square$

### A.4.4  Proof of Lemma 5

*Proof.*

$$
\begin{aligned}
&\|\mathbb{E}[\![(\boldsymbol{\beta}^T\boldsymbol{x})(\boldsymbol{\gamma}^T\boldsymbol{x})\Pi_{k=1}^K(\boldsymbol{x}^T A_k\boldsymbol{x})\boldsymbol{x}\boldsymbol{x}^T]\!]\| \\
&\geq \mathbb{E}[\![(\boldsymbol{\beta}^T\boldsymbol{x})^2(\boldsymbol{\gamma}^T\boldsymbol{x})^2\Pi_{k=1}^K(\boldsymbol{x}^T A_k\boldsymbol{x})]\!]/(\|\boldsymbol{\beta}\|\|\boldsymbol{\gamma}\|) \\
&\geq \|\boldsymbol{\beta}\|\|\boldsymbol{\gamma}\|\Pi_{k=1}^K \operatorname{tr}(A_k).
\end{aligned} \tag{59}
$$

$$
\begin{aligned}
&\left\|\mathbb{E}\big[\![(\boldsymbol{\beta}^T \boldsymbol{x})(\boldsymbol{\gamma}^T \boldsymbol{x})\Pi_{k=1}^K(\boldsymbol{x}^T A_k \boldsymbol{x})\boldsymbol{x}\boldsymbol{x}^T]\!\big]\right\| \\
&= \max_{\boldsymbol{a},\boldsymbol{b}} \mathbb{E}\big[\![(\boldsymbol{\beta}^T \boldsymbol{x})(\boldsymbol{\gamma}^T \boldsymbol{x})(\boldsymbol{a}^T \boldsymbol{x})(\boldsymbol{b}^T \boldsymbol{x})\Pi_{k=1}^K(\boldsymbol{x}^T A_k \boldsymbol{x})]\!\big]/(\|\boldsymbol{a}\|\|\boldsymbol{b}\|) \\
&\le \mathbb{E}\big[\![(\boldsymbol{a}^T \boldsymbol{x})^2(\boldsymbol{b}^T \boldsymbol{x})^2]\!\big]^{1/2}\mathbb{E}\big[\![(\boldsymbol{\beta}^T \boldsymbol{x})^2(\boldsymbol{\gamma}^T \boldsymbol{x})^2\Pi_{k=1}^K(\boldsymbol{x}^T A_k \boldsymbol{x})^2]\!\big]^{1/2}/(\|\boldsymbol{a}\|\|\boldsymbol{b}\|) \\
&\le \sqrt{3C_{2K+1}}\|\boldsymbol{\beta}\|\|\boldsymbol{\gamma}\|\Pi_{k=1}^K \operatorname{tr}(A_k)
\end{aligned}
\tag{60}
$$

$\square$

### A.4.5 Proof of Lemma 6

*Proof.* Note that the matrix $B_i$ is probably not PSD. Thus we can't apply Lemma 4 directly. But the proof is similar to that for Lemma 4.

Define

$$
m := (4P+5)^{K+2}\|\boldsymbol{\beta}\|\|\boldsymbol{\gamma}\|(\Pi_{k=1}^K \operatorname{tr}(A_k))d\log^{K+1}(n),
\tag{61}
$$

and the event, $\mathcal{E}_i := \{\|B_i\| \le m\}$. Then by Corollary 3,

$$
\mathbb{P}[\![\mathcal{E}_i]\!] \ge 1 - 2Kn^{-1}d^{-P}.
$$

Define a new random matrix $M_i := \mathbf{1}(\mathcal{E}_i)B_i$, its expectation $M := \mathbb{E}[\![M_i]\!]$ and its empirical average $\hat{M} = \frac{1}{n}\sum_{i=1}^n M_i$.

**Step 1.** By union bound, we have w.p. $1 - 2Kd^{-P}$, $M_i = B_i$ for all $i$, i.e., $\hat{M} = \hat{B}$.

**Step 2.** We now bound $\|M - B\|$, For simplicity, we replace $\boldsymbol{x}_i$ by $\boldsymbol{x}$ and $\mathcal{E}_i$ by $\mathcal{E}$.

$$
\begin{aligned}
&\|M - B\| \\
&= \|\mathbb{E}[\![B_i \mathbf{1}(\mathcal{E}_i^c)]\!]\| \\
&= \max_{\|\boldsymbol{a}\|=\|\boldsymbol{b}\|=1} \mathbb{E}\big[\![(\boldsymbol{a}^T \boldsymbol{x})(\boldsymbol{b}^T \boldsymbol{x})(\boldsymbol{\beta}^T \boldsymbol{x})(\boldsymbol{\gamma}^T \boldsymbol{x})\Pi_{k=1}^K(\boldsymbol{x}^T A_k \boldsymbol{x})\mathbf{1}(\mathcal{E}^c)]\!\big] \\
&\overset{\zeta_1}{\le} \max_{\|\boldsymbol{a}\|=\|\boldsymbol{b}\|=1} \mathbb{E}\big[\![(\boldsymbol{a}^T \boldsymbol{x})^2(\boldsymbol{b}^T \boldsymbol{x})^2(\boldsymbol{\beta}^T \boldsymbol{x})^2(\boldsymbol{\gamma}^T \boldsymbol{x})^2\Pi_{k=1}^K(\boldsymbol{x}^T A_k \boldsymbol{x})^2]\!\big]^{1/2}\mathbb{E}[\![\mathbf{1}(\mathcal{E}^c)]\!]^{1/2} \\
&= \max_{\|\boldsymbol{a}\|=\|\boldsymbol{b}\|=1} \langle \boldsymbol{a}\boldsymbol{a}^T, \mathbb{E}\big[\![(\boldsymbol{b}^T \boldsymbol{x})^2(\boldsymbol{\beta}^T \boldsymbol{x})^2(\boldsymbol{\gamma}^T \boldsymbol{x})^2\Pi_{k=1}^K(\boldsymbol{x}^T A_k \boldsymbol{x})^2\boldsymbol{x}\boldsymbol{x}^T]\!\big]\rangle^{1/2}\mathbb{E}[\![\mathbf{1}(\mathcal{E}^c)]\!]^{1/2} \\
&\overset{\zeta_2}{\le} \max_{\|\boldsymbol{a}\|=\|\boldsymbol{b}\|=1} \langle \boldsymbol{a}\boldsymbol{a}^T, C_{2K+3}\|\boldsymbol{\beta}\|^2\|\boldsymbol{\gamma}\|^2\Pi_{k=1}^K \operatorname{tr}(A_k)^2 I\rangle^{1/2}\mathbb{E}[\![\mathbf{1}(\mathcal{E}^c)]\!]^{1/2} \\
&\overset{\zeta_3}{\le} \frac{\sqrt{2KC_{2K+3}}}{\sqrt{nd^P}}\|\boldsymbol{\beta}\|\|\boldsymbol{\gamma}\|\Pi_{k=1}^K \operatorname{tr}(A_k)
\end{aligned}
\tag{62}
$$

where $\zeta_1$ is from Holder's inequality, $\zeta_2$ is because of Lemma 3 and $\zeta_3$ is because $\mathbb{E}[\![\mathbf{1}(\mathcal{E}^c)]\!] = \mathbb{P}[\![\mathcal{E}^c]\!]$.

According to Eq. (62) and Lemma 5, if $\frac{\sqrt{2KC_{2K+3}}}{\sqrt{nd^P}} \le \delta/2$, then

$$
\|M - B\| \le \frac{1}{2}\delta\|\boldsymbol{\beta}\|\|\boldsymbol{\gamma}\|\Pi_{k=1}^K \operatorname{tr}(A_k) \le \frac{1}{2}\delta\|B\|
\tag{63}
$$

Since $\delta \le 1$, we also have $\|M - B\| \le \frac{1}{2}\|B\|$, so by Lemma 5,

$$
\frac{3}{2}\|B\| \ge \|M\| \ge \frac{1}{2}\|B\| \ge \frac{1}{2}\|\boldsymbol{\beta}\|\|\boldsymbol{\gamma}\|\Pi_{k=1}^K \operatorname{tr}(A_k)
\tag{64}
$$

**Step 3.** Now we bound $\|M - \hat{M}\|$. $\|M\| \le m$ automatically holds. Since $M$ is probably not PSD, we don't have $\|\mathbb{E}[M_i^2]\| \le m\|M\|$. However, we can still show that $\mathbb{E}[M_i^2] \le O(m)\|M\|$.

$$
\begin{aligned}
&\|\mathbb{E}[M_i^2]\| \\
&\le \|\mathbb{E}[B_i^2]\| \\
&= \|\mathbb{E}[(\boldsymbol{\beta}^T \boldsymbol{x})^2 (\boldsymbol{\gamma}^T \boldsymbol{x})^2 \Pi_{k=1}^K (\boldsymbol{x}^T A_k \boldsymbol{x})^2 \|\boldsymbol{x}\|^2 \boldsymbol{x}\boldsymbol{x}^T]\| \\
&\le C_{2K+3} d \times (\|\boldsymbol{\beta}\|\|\boldsymbol{\gamma}\|\Pi_{k=1}^K \operatorname{tr}(A_k))^2 \\
&\le \frac{2C_{2K+3}}{(4P+5)^{K+2}} m\|M\|
\end{aligned}
\tag{65}
$$

We can use matrix Bernstein inequality now. Let $Z_i := M_i - M$. $\|Z_i\| \le 2m$. $\|\mathbb{E}[Z_i^2]\| \le (\frac{2C_{2K+3}}{(4P+5)^{K+2}} + 1)m\|M\|$. Define $\hat{C}_K := \frac{2C_{2K+3}}{(4P+5)^{K+2}} + 1$, then

$$
\mathbb{P}\left[\frac{1}{n}\|\sum_{i=1}^n Z_i\| \ge \delta\|M\|\right] \le 2d\exp(-\frac{\delta^2 n^2 \|M\|^2/2}{\hat{C}_K mn\|M\| + 2mn\delta\|M\|/3}) \le 2d\exp(-\frac{\delta^2 n\|M\|/2}{\hat{C}_K m + 2m\delta/3})
\tag{66}
$$

Thus, when $n \ge (P+1)(\frac{\hat{C}_K}{\delta^2} + \frac{2}{3\delta})m/\|M\|\log d$, we have w.p., $1 - c_2 d^{-P}$,

$$
\|\hat{M} - M\| \le \frac{1}{3}\delta\|M\| \le \frac{1}{2}\delta\|B\|.
$$

By Eq. (61) and Eq. (64),

$$
\begin{aligned}
(P+1)(\frac{\hat{C}_K}{\delta^2} + \frac{2}{3\delta})m/\|M\|\log d &\le c_1 \frac{\hat{C}_K}{\delta^2} \times (4P+5)^{K+3} d\log^{K+1}(n)\log(d) \\
&\le \frac{c_1}{\delta^2}(2C_{2K+3}(P+1) + (4P+5)^{K+2})d\log^{K+1}(n)\log(d)
\end{aligned}
$$

Applying the fact, $\|\hat{B} - B\| \le \|\hat{B} - \hat{M}\| + \|\hat{M} - M\| + \|M - B\|$, and Lemma 7 completes the proof. $\qquad\square$

## A.5 Proof of Lemma 7

*Proof.* Assume we require $n \geq cd\log(d)\log^{K+1}(n)$ and we have $n \geq bcd\log(d)\log^A(d)$, where $b, A$ depends only on $K$.

$$n \geq cd\log d\log^{K+1}(n)$$

$$\Uparrow$$

$$\frac{n}{\log^{K+1}(n)} \geq cd\log d$$

$$\Uparrow$$

$$\frac{bcd\log(d)\log^A(d)}{\log^{K+1}(bcd\log(d)\log^A(d))} \geq cd\log d$$

$$\Uparrow$$

$$b\log^A(d) \geq \log^{K+1}(bcd\log(d)\log^A(d))$$

$$\Uparrow$$

$$\log b + A\log\log(d) \geq (K+1)\log(\log(b)+\log(c)+\log(d)+(A+1)\log\log(d))$$

$$\Uparrow$$

$$\log b + A\log\log(d) \geq (K+1)\log(4\max\{\log(b),\log(c),\log(d),(A+1)\log\log(d)\}) \tag{67}$$

$$\Uparrow$$

$$\begin{cases} \log b \geq (K+1)\log(4\log(b)) \\ \log b \geq (K+1)\log(4\log(c)) \\ \log b + A\log\log(d) \geq (K+1)\log(4\log(d)) \\ \log b + A\log\log(d) \geq (K+1)\log(4(A+1)\log\log(d)) \end{cases}$$

$$\Uparrow$$

$$\begin{cases} b \geq K^{4K} \\ b \geq 4^{K+1}\log^{K+1}(c) \\ A \geq K+1 \\ b \geq (4(A+1))^{K+1} \end{cases}$$

$$\Uparrow$$

$$\begin{cases} b = K^{4K}\log^{K+1}(c) \\ A = K+1 \end{cases}$$

$\square$

# B   Proofs of Tensor Method for Initialization

## B.1   Some Lemmata

We will use the following lemma to guarantee the robust tensor power method. The proofs of these lemmata will be found in Sec. B.4.

**Lemma 8** ( Some properties of thrid-order tensor). *If $T \in \mathbb{R}^{d\times d\times d}$ is a supersymmetric tensor, i.e.,$T_{ijk}$ is equivalent for any permutation of the index, then the operator norm defined as*

$$\|T\|_{op} := \sup_{\|\boldsymbol{a}\|=1}|T(\boldsymbol{a},\boldsymbol{a},\boldsymbol{a})|$$

**Property 1.** $\|T\|_{op} = \sup_{\|\boldsymbol{a}\|=\|\boldsymbol{b}\|=\|\boldsymbol{c}\|=1}|T(\boldsymbol{a},\boldsymbol{b},\boldsymbol{c})|$

**Property 2.** $\|T\|_{op} \leq \|T_{(1)}\| \leq \sqrt{K}\|T\|_{op}$

**Property 3.** *If $T$ is a rank-one tensor, then $\|T_{(1)}\| = \|T\|_{op}$*

**Property 4.** *For any matrix $W \in \mathbb{R}^{d\times d'}$, $\|T(W,W,W)\|_{op} \leq \|T\|_{op}\|W\|^3$*

**Lemma 9** (Approximation error for the second moment). *Let $\{\boldsymbol{x}_i, y_i\}_{i\in[n]}$ be generated from the mixed linear regression model* (3). *Define $M_2 := \sum_{k=[K]}2p_k\boldsymbol{w}_k^*\otimes\boldsymbol{w}_k^*$ and $\hat{M}_2 := \frac{1}{n}\sum_{i\in[n]}y_i^2(\boldsymbol{x}_i\otimes$*

$\boldsymbol{x}_i - I$). Then with $n \geq c_1 \frac{1}{p_{min}\delta_2^2} d \log^2(d)$, we have w.p. $1 - c_2 K d^{-2}$,

$$\|\hat{M}_2 - M_2\| \leq \delta_2 \sum_k p_k \|\boldsymbol{w}_k^*\|^2 \tag{68}$$

where $c_1, c_2$ are universal constants.

And for any fixed orthogonal matrix $Y \in \mathbb{R}^{d \times K}$, with the same condition, we have

$$\|Y^T(\hat{M}_2 - M_2)Y\| \leq \delta_2 \sum_k p_k \|\boldsymbol{w}_k^*\|^2 \tag{69}$$

**Lemma 10** (Subspace Estimation). *Let $M_2, M_3$ be*

$$M_2 = \sum_{k=[K]} 2p_k \boldsymbol{w}_k^* \otimes \boldsymbol{w}_k^*, \text{ and } M_3 = \sum_{k=[K]} 6p_k \boldsymbol{w}_k^* \otimes \boldsymbol{w}_k^* \otimes \boldsymbol{w}_k^*, \tag{70}$$

*and $\hat{M}_2$ be an estimate of $M_2$. Assume $\|\hat{M}_2 - M_2\| \leq \delta \sigma_K(M_2)$ and $\delta \leq \frac{1}{6}$. Let $Y$ be the returned matrix of the power method after $O(\log(1/\delta))$ steps. Define $R_2 = Y^T M_2 Y$ and $R_3 = M_3(Y, Y, Y)$. Then $\|R_2\| \leq \|M_2\|$ and $\|R_3\|_{op} \leq \|M_3\|_{op}$. We also have*

$$\|YY^T \boldsymbol{w}_k^* - \boldsymbol{w}_k^*\| \leq 3\delta \|\boldsymbol{w}_k^*\|, \forall k \tag{71}$$

*and*

$$\sigma_K(R_2) \geq \frac{3}{4}\sigma_K(M_2)$$

**Lemma 11** (Approximation error for the third moment). *Let $\{\boldsymbol{x}_i, y_i\}_{i \in [n]}$ be drawn from the mixed linear regression model (3). Let $Y \in \mathbb{R}^{d \times K}$ be any fixed orthogonal matrix that satisfies, $\|YY^T \boldsymbol{w}_k^* - \boldsymbol{w}_k^*\| \leq \frac{1}{2}\|\boldsymbol{w}_k^*\|, \forall k$, and $\boldsymbol{r}_i = Y^T \boldsymbol{x}_i$, for all $i \in [n]$. Let*

$$\hat{R}_3 = \frac{1}{n}\sum_{i \in [n]} y_i^3 (\boldsymbol{r}_i \otimes \boldsymbol{r}_i \otimes \boldsymbol{r}_i - \sum_{j \in [K]} \boldsymbol{e}_j \otimes \boldsymbol{r}_i \otimes \boldsymbol{e}_j - \sum_{j \in [K]} \boldsymbol{e}_j \otimes \boldsymbol{e}_j \otimes \boldsymbol{r}_i - \sum_{j \in [K]} \boldsymbol{r}_i \otimes \boldsymbol{e}_j \otimes \boldsymbol{e}_j)$$

*and*

$$R_3 = \sum_{k=[K]} 6p_k (Y^T \boldsymbol{w}_k^*) \otimes (Y^T \boldsymbol{w}_k^*) \otimes (Y^T \boldsymbol{w}_k^*)$$

*Then if $n \geq c_3 \frac{1}{p_{min}\delta_3^2} K^3 \log^4(d)$ and $3\sqrt{C_5} n^{-1/2} d^{-1} \leq \frac{\delta_3}{4}$, we have w.p. $1 - c_4 K d^{-2}$*

$$\|\hat{R}_3 - R_3\|_{op} \leq \delta_3 \sum_{k \in [K]} p_k \|\boldsymbol{w}_k^*\|^3,$$

*where $c_3$ and $c_4$ are universal constant.*

**Lemma 12** (Robust Tensor Power Method. Similar to Lemma 4 in [7]). *Let $R_2 = \sum_{k=1}^K p_k \boldsymbol{u}_k \otimes \boldsymbol{u}_k$ and $R_3 = \sum_{k=1}^K p_k \boldsymbol{u}_k \otimes \boldsymbol{u}_k \otimes \boldsymbol{u}_k$, where $\boldsymbol{u}_k \in \mathbb{R}^K$ can be any fixed vector. Define $\sigma_K := \sigma_K(R_2)$. Assume the estimations of $R_2$ and $R_3$, $\hat{R}_2$ and $\hat{R}_3$ respectively, satisfy $\|R_2 - \hat{R}_2\|_{op} \leq \epsilon_2$ and $\|R_3 - \hat{R}_3\|_{op} \leq \epsilon_3$ with*

$$\epsilon_2 \leq \sigma_K/3, \ 8\|R_3\|_{op}\sigma_K^{-5/2}\epsilon_2 + 2\sqrt{2}\sigma_K^{-3/2}\epsilon_3 \leq c_T \frac{1}{K\sqrt{p_{max}}}, \tag{72}$$

*for some constant $c_T$. Let the whitening matrix $\hat{W} = \hat{U}_2 \hat{\Lambda}_2^{-1/2} \hat{U}_2^T$, where $\hat{R}_2 = \hat{U}_2 \hat{\Lambda}_2 \hat{U}_2^T$ is the eigendecomposition of $\hat{R}_2$. Then w.p. $1 - \eta$, the eigenvalues $\{\hat{a}_k\}_{k=1}^K$ and the eigenvectors $\{\hat{\boldsymbol{v}}_k\}_{k=1}^K$ computed from the whitened tensor $\hat{R}_3(\hat{W}, \hat{W}, \hat{W}) \in \mathbb{R}^{K \times K \times K}$ by using the robust tensor power method [2] will satisfy*

$$\|(\hat{W}^T)^\dagger(\hat{a}_k \hat{\boldsymbol{v}}_k) - \boldsymbol{u}_k\| \leq \kappa_2 \epsilon_2 + \kappa_3 \epsilon_3$$

*where $\kappa_2 = 3\|R_2\|^{1/2}\sigma_K^{-1} + 200\|R_2\|^{1/2}\|R_3\|_{op}\sigma_K^{-5/2}$, $\kappa_3 = 75\|R_2\|^{1/2}\sigma_K^{-3/2}$ and $\eta$ is related to the computational time by $O(\log(1/\eta))$.*

**Remark:** This lemma differs from Lemma 4 of [7] in the requirement on $\epsilon_2, \epsilon_3$. Lemma 4 in [7] treats $\epsilon_2, \epsilon_3$ in the same order (that are bounded by the same value), however, they should have different order because one is for second-order moments and the other is for third-order moments.

## B.2 Proof of Theorem 3

**Proof Details.** We state the proof outline here,

1. $\|\hat{M}_2 - M_2\| \le \epsilon_{M_2}$ by Matrix Bernstein's inequality.
2. $\|YY^T \boldsymbol{w}_k^* - \boldsymbol{w}_k^*\| \le \epsilon_Y \|\boldsymbol{w}_k^*\|$ for all $k \in [K]$ by Davis-Kahan's theorem [10].
3. $\|\hat{R}_2 - R_2\| \le \epsilon_2$ by Matrix Bernstein's inequality.
4. $\|\hat{R}_3 - R_3\|_{op} \le \epsilon_3$ by Matrix Bernstein's inequality after matricizing tensor.
5. Let $\hat{\boldsymbol{u}}_k = (\hat{W}^T)^\dagger (\hat{a}_k \hat{\boldsymbol{v}}_k)$. Then $\|\hat{\boldsymbol{u}}_k - Y^T \boldsymbol{w}_k^*\| \le \epsilon_{\boldsymbol{u}}$ by the robust tensor power method.
6. Finally, $\|\boldsymbol{w}_k^{(0)} - \boldsymbol{w}_k^*\| \le c_6 \Delta_{min}$ by combining the results of Step 2 and Step 5.

The lemmata in Appendix B.1 provide the bound for the above steps: Lemma 9 for Step 1, Lemma 10 for Step 2 and Step 3, Lemma 11 for Step 4, and Lemma 12 for Step 5. Now we show the details. Define

$$\bar{\kappa}_2 := 4\|M_2\|^{1/2}\sigma_K^{-1}(M_2) + 412\|M_2\|^{1/2}\|M_3\|_{op}\sigma_K^{-5/2}(M_2)$$

and

$$\bar{\kappa}_3 := 116\|M_2\|^{1/2}\sigma_K^{-3/2}(M_2).$$

By Lemma 10, we have $\bar{\kappa}_3 \ge \kappa_3$ and $\bar{\kappa}_2 \ge \kappa_2$ for any orthogonal matrix $Y$.

$$
\begin{aligned}
\|\boldsymbol{w}_k^{(0)} - \boldsymbol{w}_k^*\| &\overset{\xi_1}{\le} \|Y\hat{\boldsymbol{u}}_k - YY^T \boldsymbol{w}_k^*\| + \|YY^T\boldsymbol{w}_k^* - \boldsymbol{w}_k^*\| \\
&\overset{\xi_2}{\le} \bar{\kappa}_2\|\hat{R}_2 - R_2\| + \bar{\kappa}_3\|\hat{R}_3 - R_3\|_{op} + \frac{2}{3}\delta_{M_2}\|\boldsymbol{w}_k^*\|\sigma_K^{-1}(M_2)\sum_k p_k\|\boldsymbol{w}_k^*\|^2 \\
&\overset{\xi_3}{\le} \bar{\kappa}_2\delta_2 \sum_k p_k\|\boldsymbol{w}_k^*\|^2 + \bar{\kappa}_3\delta_3 \sum_k p_k\|\boldsymbol{w}_k^*\|^3 + \frac{2}{3}\delta_{M_2}\sigma_K^{-1}(M_2)(\max_k\|\boldsymbol{w}_k^*\|)\sum_k p_k\|\boldsymbol{w}_k^*\|^2
\end{aligned}
$$
(73)

where $\xi_1$ is due to triangle inequality, $\xi_2$ is due to Lemma 12, Lemma 10 and Lemma 9, and $\xi_3$ is due to Lemma 9 and Lemma 11. Therefore, we can set

$$\delta_2 \le \frac{c_6\Delta_{min}}{3\bar{\kappa}_2 \sum_k p_k\|\boldsymbol{w}_k^*\|^2},$$

$$\delta_3 \le \frac{c_6\Delta_{min}}{3\bar{\kappa}_3 \sum_{k\in[K]} p_k\|\boldsymbol{w}_k^*\|^3}$$

and

$$\delta_{M_2} \le \frac{c_6\Delta_{min}}{2\sigma_K^{-1}(M_2)(\max_k\|\boldsymbol{w}_k^*\|)\sum_k p_k\|\boldsymbol{w}_k^*\|^2},$$

such that $\|\boldsymbol{w}_k^{(0)} - \boldsymbol{w}_k^*\| \le c_6\Delta_{min}$. Note that Lemma 12 also requires Eq. (72), which can be satisfied if

$$\|\hat{R}_2 - R_2\| \le \min\{\frac{\sigma_K(M_2)}{4}, \frac{c_T\sigma_K(M_2)^{5/2}}{34\|M_3\|_{op}K\sqrt{p_{max}}}\}$$

and

$$\|\hat{R}_3 - R_3\|_{op} \le \frac{c_T\sigma_K(M_2)^{3/2}}{6K\sqrt{p_{max}}}.$$

Therefore, we require

$$\delta_2 \le \delta_2^* := \frac{1}{\sum_k p_k\|\boldsymbol{w}_k^*\|^2}\min\{\frac{\sigma_K(M_2)}{4}, \frac{c_T\sigma_K(M_2)^{5/2}}{34\|M_3\|_{op}K\sqrt{p_{max}}}, \frac{c_6\Delta_{min}}{3\bar{\kappa}_2}\}$$

$$\delta_3 \le \delta_3^* := \frac{1}{\sum_{k\in[K]} p_k\|\boldsymbol{w}_k^*\|^3}\min\{\frac{c_6\Delta_{min}}{3\bar{\kappa}_3}, \frac{c_T\sigma_K(M_2)^{3/2}}{6K\sqrt{p_{max}}}\}$$

$$\delta_{M_2} \le \delta_{M_2}^* := \frac{c_6\Delta_{min}}{2\sigma_K^{-1}(M_2)(\max_k\|\boldsymbol{w}_k^*\|)\sum_k p_k\|\boldsymbol{w}_k^*\|^2},$$

Now we analyze the sample complexity. $\delta^*_{M_2}, \delta^*_2, \delta^*_3$ correspond to the sample sets, $\Omega_{M_2}$, $\Omega_2$ and $\Omega_3$ respectively. By Lemma 9, Lemma 11, we require

$$|\Omega_{M_2}| \geq c_{M_2} \frac{1}{p_{min}\delta^{*2}_{M_2}} d \log^2(d)$$

$$|\Omega_2| \geq c_2 \frac{1}{p_{min}\delta^{*2}_2} d \log^2(d)$$

$$|\Omega_3| \geq c_3 \frac{1}{p_{min}\delta^{*2}_3} K^3 \log^{11/2}(d),$$

and $3\sqrt{C_5}n^{-1/2}d^{-1} \leq \frac{\delta_3}{4}$. For the probability, we can set $\eta = d^{-2}$ in Lemma 12 by scarifying a little more computational time, which is in the order of $O(\log(d))$. Therefore, the final probability is at least $1 - O(Kd^{-2})$.

### B.3 Proof of Theorem 4

According to Theorem 2, after $T_0 = O(\log d)$ iterations, we arrive the local convexity region in Corollary 1. Then we just need one more set of samples, but still need $O(\log(1/\epsilon))$ iterations to achieve $1/\epsilon$ precision. By Theorem 1, Corollary 1, Theorem 2 and Theorem 3, we can partition the dataset into $|\Omega^{(t)}| = O(d(K\log(d))^{2K+2})$ for all $t = 0, 1, 2, \cdots, T_0 + 1$ to satisfy their sample complexity requirement. This complete the proof.

### B.4 Proofs of Some Lemmata

#### B.4.1 Proof of Lemma 8

*Proof.* **Property 1.** See the proof in Lemma 21 of [19].

**Property 2.**

$$\|T_{(1)}\| = \max_{\|\boldsymbol{a}\|=1} \|T(\boldsymbol{a}, I, I)\|_F \leq \max_{\|\boldsymbol{a}\|=1} \sqrt{K}\|T(\boldsymbol{a}, I, I)\| = \max_{\|\boldsymbol{a}\|=\|\boldsymbol{b}\|=1} \sqrt{K}|T(\boldsymbol{a}, \boldsymbol{b}, \boldsymbol{b})| = \|T\|_{op}.$$

Obviously, $\max_{\|\boldsymbol{a}\|=1} \|T(\boldsymbol{a}, I, I)\|_F \geq \|T\|_{op}$.
**Property 3.** Let $T = \boldsymbol{v} \otimes \boldsymbol{v} \otimes \boldsymbol{v}$.

$$\|T_{(1)}\| = \max_{\|\boldsymbol{a}\|=1} \|T(\boldsymbol{a}, I, I)\|_F = \max_{\|\boldsymbol{a}\|=1} \|\boldsymbol{v}\|^2 (\boldsymbol{v}^T \boldsymbol{a})^2 = \|\boldsymbol{v}\|^3 = \max_{\|\boldsymbol{a}\|=1} |(\boldsymbol{v}^T \boldsymbol{a})^3| = \|T\|_{op}.$$

**Property 4.** There exists a $\boldsymbol{u} \in \mathbb{R}^{d'}$ with $\|\boldsymbol{u}\| = 1$ such that

$$\|T(W, W, W)\|_{op} = |T(W\boldsymbol{u}, W\boldsymbol{u}, W\boldsymbol{u})| \leq \|T\|_{op}\|W\boldsymbol{u}\|^3 \leq \|T\|_{op}\|W\|^3$$

$\square$

#### B.4.2 Proof of Lemma 9

*Proof.* Define $M_2^{(k)} := 2\boldsymbol{w}_k^* \boldsymbol{w}_k^{*T}$ and $\hat{M}_2^{(k)} = \frac{1}{|S_k|} \sum_{i \in S_k} y_i^2 (\boldsymbol{x}_i \otimes \boldsymbol{x}_i - I)$, where $S_k \subset [n]$ is the index set for samples from the $k$-th model. Since we assume $|S_k| = p_k n$, $\hat{M}_2 = \sum_{k \in [K]} p_k M_2^{(k)}$. We first bound $\|\hat{M}_2^{(k)} - M_2^{(k)}\|$. By Lemma 4 with $K = 1$, $A_1 = \boldsymbol{w}_k^* \boldsymbol{w}_k^{*T}$, then if $|S_k| \geq c_1 \frac{1}{\delta^2} d \log^2(d)$, we have w.p., $1 - c_2 d^{-2}$,

$$\|\frac{1}{|S_k|} \sum_{i \in S_k} y_i^2 \boldsymbol{x}_i \boldsymbol{x}_i^T - \|\boldsymbol{w}_k^*\|^2 I - 2\boldsymbol{w}_k^* \boldsymbol{w}_k^*\| \leq \delta\|\boldsymbol{w}_k^*\|^2.$$

By Lemma 4 with $K = 0$, we have w.p. at least $1 - d^{-2}$,

$$\|\frac{1}{|S_k|} \sum_{i \in S_k} \boldsymbol{x}_i \boldsymbol{x}_i^T - I\| \leq \delta$$

Then

$$\|\frac{1}{|S_k|}\sum_{i\in S_k}(\boldsymbol{x}_i^T\boldsymbol{w}_k^*)^2 - \|\boldsymbol{w}_k^*\|^2\| \leq \|\frac{1}{|S_k|}\sum_{i\in S_k}\boldsymbol{x}_i\boldsymbol{x}_i^T - I\|\|\boldsymbol{w}_k^*\|^2 \leq \delta\|\boldsymbol{w}_k^*\|^2.$$

Thus

$$\|\frac{1}{|S_k|}\sum_{i\in S_k}y_i^2(\boldsymbol{x}_i\boldsymbol{x}_i^T - I) - 2\boldsymbol{w}_k^*\boldsymbol{w}_k^*\| \leq 2\delta\|\boldsymbol{w}_k^*\|^2.$$

And w.p. $1 - O(Kd^{-2})$,

$$\|\hat{M}_2 - M_2\| \leq 2\delta\sum_k p_k\|\boldsymbol{w}_k^*\|^2.$$

$\square$

### B.4.3 Proof of Lemma 10

*Proof.* $\|R_2\| \leq \|Y\|^2\|M_2\| = \|M_2\|$. By Property 4 in Lemma 8, $\|R_3\|_{op} \leq \|Y\|^3\|M_3\|_{op} = \|M_3\|_{op}$. Let $U$ be the top-$K$ eigenvectors of $M_2$. Then $U = \text{span}(\boldsymbol{w}_1^*, \boldsymbol{w}_2^*, \cdots, \boldsymbol{w}_K^*)$. Let $\bar{Y} \in \mathbb{R}^{d\times K}$ be the top-$K$ eigenvectors of $\hat{M}_2$. By Lemma 9 in [19] (Davis-Kahan's theorem [10] can also prove it),

$$\|(I - \bar{Y}\bar{Y}^T)UU^T\| \leq \frac{3}{2}\delta.$$

According to Theorem 7.2 in [3], after $t$ steps of the power method, we have

$$\|\bar{Y}\bar{Y}^T - Y^{(t)}Y^{(t)T}\| \leq (\frac{\sigma_{K+1}(\hat{M}_2)}{\sigma_K(\hat{M}_2)})^t\|\bar{Y}\bar{Y}^T - Y^{(0)}Y^{(0)T}\|.$$

When $\delta \leq 1/3$, by Weyl's inequality, we have $\sigma_{K+1}(\hat{M}_2) \leq \frac{1}{3}\sigma_K(M_2)$ and $\sigma_K(\hat{M}_2) \geq \frac{2}{3}\sigma_K(M_2)$. Therefore, after $t = \log(2/(3\delta))$ steps of the power method, we have

$$\|\bar{Y}\bar{Y}^T - Y^{(t)}Y^{(t)T}\| \leq \frac{3}{2}\delta$$

Let $Y = Y^{(t)}$. We have

$$\|YY^T - UU^T\| \leq \|YY^T - \bar{Y}\bar{Y}^T\| + \|UU^T - \bar{Y}\bar{Y}^T\| \leq 3\delta$$

and

$$\|YY^T\boldsymbol{w}_k^* - \boldsymbol{w}_k^*\| \leq \|YY^T - UU^T\|\|\boldsymbol{w}_k^*\| \leq 3\delta\|\boldsymbol{w}_k^*\|$$

Now we consider $\sigma_K(R_2)$. The proof is similar to that for Property 3 in Lemma 9 in [19].

$$\sigma_K(R_2) \geq \sigma_K(M_2)\sigma_K^2(Y^TU)$$

Note that $\|Y_\perp^TU\| = \|YY^T - UU^T\|$, where $Y_\perp$ is the subspace orthogonal to $Y$. For any normalized vector $\boldsymbol{v}$,

$$\|Y^TU\boldsymbol{v}\|^2 = \|U\boldsymbol{v}\|^2 - \|Y_\perp^TU\boldsymbol{v}\|^2 \geq 1 - (3\delta)^2 \geq \frac{3}{4}$$

Therefore, we have $\sigma_K(R_2) \geq \frac{3}{4}\sigma_K(M_2)$. $\square$

### B.4.4 Proof of Lemma 11

*Proof.* We prove it by matricizing the tensor. Define

$$G_i = y_i^3(\boldsymbol{r}_i\otimes\boldsymbol{r}_i\otimes\boldsymbol{r}_i - \sum_{j\in[K]}\boldsymbol{e}_j\otimes\boldsymbol{r}_i\otimes\boldsymbol{e}_j - \sum_{j\in[K]}\boldsymbol{e}_j\otimes\boldsymbol{e}_j\otimes\boldsymbol{r}_i - \sum_{j\in[K]}\boldsymbol{r}_i\otimes\boldsymbol{e}_j\otimes\boldsymbol{e}_j).$$

Like in Lemma 9, we first bound $\|\hat{R}_3^{(k)} - R_3^{(k)}\|_{op}$, where $\hat{R}_3^{(k)} = \frac{1}{|S_k|}\sum_{i\in S_k}G_i$, and $R_3^{(k)} = 6(Y^T\boldsymbol{w}_k^*)\otimes(Y^T\boldsymbol{w}_k^*)\otimes(Y^T\boldsymbol{w}_k^*)$.

$$\|R_3^{(k)}\|_{op} = 6\|Y^T\boldsymbol{w}_k^*\|^3.$$

By Lemma 10, $\frac{1}{2}\|\boldsymbol{w}_k^*\| \le \|Y^T\boldsymbol{w}_k^*\| \le \frac{3}{2}\|\boldsymbol{w}_k^*\|$. Thus

$$\frac{3}{4}\|\boldsymbol{w}_k^*\|^3 \le \|R_3^{(k)}\|_{op} \le \frac{81}{4}\|\boldsymbol{w}_k^*\|^3. \tag{74}$$

Then

$$\|G_i\|_{op} \le 4|\boldsymbol{x}_i^T\boldsymbol{w}_k^*|^3\|\boldsymbol{r}_i\|^3, \tag{75}$$

By Corollary 4, we have w.p., $1 - n^{-1}d^{-2}$, $\|\boldsymbol{r}_i\|^2 \le 4K\log n$. Thus, w.p. $1 - 4n^{-1}d^{-2}$,

$$\|G_i\|_{op} \le 4 \times 12^{3/2}\|\boldsymbol{w}_k^*\|^3\log^3(n)(4K)^{3/2}$$

Define $m := c_6\|\boldsymbol{w}_k^*\|^3 K^{3/2}\log^3(n)$ for constant $c_6 = 4 \times (48)^{3/2}$, and the event

$$\mathcal{E}_i := \{\|G_i\|_{op} \le m\}$$

Then $\mathbb{P}[\mathcal{E}_i^c] \le 4n^{-1}d^{-2}$. Define a new tensor $B_i = \mathbf{1}(\mathcal{E}_i)G_i$, its expectation $B = \mathbb{E}[B_i]$ (the expectation is over all samples from the $k$-th components) and its empirical average $\hat{B} = \frac{1}{|S_k|}\sum_{i\in[S_k]} B_i$.

**Step 1.** So we have $B_i = G_i$ for all $i \in S_k$ w.p. $1 - 4d^{-2}$, i.e.,

$$\hat{R}_3^{(k)} = \hat{B} \tag{76}$$

**Step 2.** We bound $\|B - R_3^{(k)}\|_{op}$

$$\begin{aligned}
\|B - R_3^{(k)}\|_{op} &= \|\mathbb{E}[\mathbf{1}(\mathcal{E}_i^c)G_i]\|_{op} \\
&= \max_{\|\boldsymbol{a}\|=1}|\mathbb{E}[\mathbf{1}(\mathcal{E}_i^c)G_i(\boldsymbol{a},\boldsymbol{a},\boldsymbol{a})]| \\
&\le \mathbb{E}[\mathbf{1}(\mathcal{E}_i^c)]^{1/2}\max_{\|\boldsymbol{a}\|=1}|\mathbb{E}[G_i(\boldsymbol{a},\boldsymbol{a},\boldsymbol{a})^2]|^{1/2} \\
&\le 2n^{-1/2}d^{-1}\max_{\|\boldsymbol{a}\|=1}|\mathbb{E}[(y_i^3((\boldsymbol{r}_i^T\boldsymbol{a})^3 - 3\boldsymbol{r}_i^T\boldsymbol{a}))^2]|^{1/2} \\
&\le 2n^{-1/2}d^{-1}\max_{\|\boldsymbol{a}\|=1}|\mathbb{E}[(\boldsymbol{w}_k^{*T}\boldsymbol{x}_i)^6((\boldsymbol{x}_i^TY\boldsymbol{a})^6 + 9(\boldsymbol{x}_i^TY\boldsymbol{a})^2)]|^{1/2} \\
&\le 2n^{-1/2}d^{-1}\sqrt{2C_5}\|\boldsymbol{w}_k^*\|^3 \\
&\overset{\xi}{\le} 3\sqrt{C_5}n^{-1/2}d^{-1}\|R_3^{(k)}\|_{op},
\end{aligned} \tag{77}$$

where $\xi$ is due to Eq. (74). Therefore, if $3\sqrt{C_5}n^{-1/2}d^{-1} \le \frac{\delta_3}{4}$, we have

$$\|B - R_3^{(k)}\|_{op} \le \frac{3\delta_3}{8}\|\boldsymbol{w}_k^*\|^3 \le \frac{\delta_3}{2}\|R_3^{(k)}\|_{op} \tag{78}$$

And further if $\delta_3 \le 1$, combining Eq. (74),

$$\frac{3}{8}\|\boldsymbol{w}_k^*\|^3 \le \frac{1}{2}\|R_3^{(k)}\|_{op} \le \|B\|_{op} \le \frac{3}{2}\|R_3^{(k)}\|_{op} \le 32\|\boldsymbol{w}_k^*\|^3$$

**Step 3.** We bound $\|\hat{B} - B\|_{op}$. Let $Z_i = (B_i - B)_{(1)}$.

$$\begin{aligned}
\|B_{(1)}\| &\le \max_{\|\boldsymbol{a}\|=1}\|B_{(1)}\boldsymbol{a}\| \\
&= \max_{\|\boldsymbol{a}\|=1}\|B(\boldsymbol{a}, I, I)\|_F \\
&\le \max_{\|\boldsymbol{a}\|=1}K\|B(\boldsymbol{a}, I, I)\| \\
&\le \max_{\|\boldsymbol{a}\|=1}\max_{\|\boldsymbol{b}\|=1}\sqrt{K}|B(\boldsymbol{a}, \boldsymbol{b}, \boldsymbol{b})| \\
&\overset{\xi}{=} \sqrt{K}\|B\|_{op} \\
&\le 32\sqrt{K}\|\boldsymbol{w}_k^*\|^3
\end{aligned} \tag{79}$$

where $\xi$ is due to Lemma 8.

$$\|Z_i\| \le \|B_{i(1)}\| + \|B_{(1)}\| \le \sqrt{K}(\|B_i\|_{op} + \|B\|_{op}) \le 2\sqrt{K}m$$

Now consider $\|\mathbb{E}[\![Z_i Z_i^T]\!]\|$ and $\|\mathbb{E}[\![Z_i^T Z_i]\!]\|$.

$$\mathbb{E}[\![Z_i Z_i^T]\!] = \mathbb{E}[\![(B_{i(1)} - B_{(1)})(B_{i(1)} - B_{(1)})^T]\!] = \mathbb{E}[\![B_{i(1)}B_{i(1)}^T]\!] - B_{(1)}B_{(1)}^T$$

$$
\begin{aligned}
\|\mathbb{E}[\![B_{i(1)}B_{i(1)}^T]\!]\| &\le \|\mathbb{E}[\![G_{i(1)}G_{i(1)}^T]\!]\| \\
&\le \|\mathbb{E}[\![(\boldsymbol{w}_k^{*T}\boldsymbol{x})^6(\|\boldsymbol{r}\|^4\boldsymbol{r}\boldsymbol{r}^T + 2\|\boldsymbol{r}\|^2 I + (K+6)\boldsymbol{r}\boldsymbol{r}^T - 6\|\boldsymbol{r}\|^2\boldsymbol{r}\boldsymbol{r}^T)]\!]\| \\
&\le \|\mathbb{E}[\![(\boldsymbol{w}_k^{*T}\boldsymbol{x})^6(\|Y^T\boldsymbol{x}\|^4 Y^T\boldsymbol{x}\boldsymbol{x}^T Y + 2\|Y^T\boldsymbol{x}\|^2 I + (K+6)Y^T\boldsymbol{x}\boldsymbol{x}^T Y)]\!]\| \\
&\le 2C_5 K^2\|\boldsymbol{w}_k^*\|^6,
\end{aligned}
\tag{80}
$$

where the last inequality is due to Lemma 3. Thus

$$\|\mathbb{E}[\![Z_i Z_i^T]\!]\| \le 3C_5 K^2\|\boldsymbol{w}_k^*\|^6$$

Similarly $\mathbb{E}[\![Z_i^T Z_i]\!] = \mathbb{E}[\![B_{i(1)}^T B_{i(1)}]\!] - B_{(1)}^T B_{(1)}$ and $\|B_{(1)}^T B_{(1)}\| \le \|B_{(1)}\|^2$.

$$
\begin{aligned}
&\|\mathbb{E}[\![B_{i(1)}^T B_{i(1)}]\!]\| \\
&\le \|\mathbb{E}[\![G_{i(1)}^T G_{i(1)}]\!]\| \\
&\le \max_{\|A\|_F=1, A \text{ sym.}} \mathbb{E}[\![y_i^6\|\boldsymbol{r}^T A\boldsymbol{r} - (2A\boldsymbol{r} + \operatorname{tr}(A)\boldsymbol{r})\|^2]\!] \\
&\le \max_{\|A\|_F=1, A \text{ sym.}} \mathbb{E}[\![(\boldsymbol{w}_k^{*T}\boldsymbol{x})^6((\boldsymbol{r}^T A\boldsymbol{r})^2\|\boldsymbol{r}\|^2 + 4\boldsymbol{r}^T A^2\boldsymbol{r} + \operatorname{tr}^2(A)\|\boldsymbol{r}\|^2 + |\operatorname{tr}(A)\boldsymbol{r}^T A\boldsymbol{r}|(4 + 2\|\boldsymbol{r}\|^2))]\!] \\
&\le \max_{\|A\|_F=1, A \text{ sym.}} \mathbb{E}[\![(\boldsymbol{w}_k^{*T}\boldsymbol{x})^6(\|\boldsymbol{r}\|^6\|A\|_F^2 + 4\|\boldsymbol{r}\|^2\operatorname{tr}(A^2) + \operatorname{tr}^2(A)\|\boldsymbol{r}\|^2 + (4 + 2\|\boldsymbol{r}\|^2)\|\boldsymbol{r}\|^2|\operatorname{tr}(A)|\|A\|_F)]\!] \\
&\le \mathbb{E}[\![(\boldsymbol{w}_k^{*T}\boldsymbol{x})^6(\|\boldsymbol{r}\|^6 + 4\|\boldsymbol{r}\|^2 + K\|\boldsymbol{r}\|^2 + \sqrt{K}(4 + 2\|\boldsymbol{r}\|^2)\|\boldsymbol{r}\|^2)]\!] \\
&= \mathbb{E}[\![(\boldsymbol{w}_k^{*T}\boldsymbol{x})^6(\|Y^T\boldsymbol{x}\|^6 + 2\sqrt{K}\|Y^T\boldsymbol{x}\|^4 + 4(\sqrt{K}+1)\|Y^T\boldsymbol{x}\|^2 + K\|Y^T\boldsymbol{x}\|^2)]\!] \\
&\le 2C_5 K^3\|\boldsymbol{w}_k^*\|^6
\end{aligned}
\tag{81}
$$

Therefore,

$$\|\mathbb{E}[\![Z_i^T Z_i]\!]\| \le 3C_5 K^3\|\boldsymbol{w}_k^*\|^6,$$

and

$$\max\{\|\mathbb{E}[\![Z_i^T Z_i]\!]\|, \|\mathbb{E}[\![Z_i Z_i^T]\!]\|\} \le 3C_5 K^3\|\boldsymbol{w}_k^*\|^6 \le c_{m2}K^{3/2}m\|\boldsymbol{w}_k^*\|^3$$

Now we are ready to apply matrix Bernstein's inequality.

$$\mathbb{P}\left[\!\!\left[\frac{1}{|S_k|}\|\sum_{i \in S_k} Z_i\| \ge t\right]\!\!\right] \le 2K^2 \exp\left(-\frac{-|S_k|t^2/2}{c_{m2}K^{3/2}m\|\boldsymbol{w}_k^*\|^3 + 2\sqrt{K}mt/3}\right)
\tag{82}$$

Setting $t = \delta_3\|\boldsymbol{w}_k^*\|^3$, we have when

$$|S_k| \ge \hat{c}_3\frac{1}{\delta_3^2}K^3\log^3(n)\log(d)
\tag{83}$$

w.p. $1 - d^{-2}$,

$$\|\hat{B} - B\|_{op} \le \|\frac{1}{|S_k|}\sum_{i \in S_k} Z_i\| \le \delta_3\|\boldsymbol{w}_k^*\|^3,
\tag{84}$$

for some universal constant $\hat{c}_3$. And there exists some constant $c_3$, such that $|S_k| \geq \hat{c}_3 \frac{1}{\delta_3^2} K^3 \log^4(d)$ will imply (83). **Step 4.** Combing all the $K$ components. With above three steps for $k$-th component, i.e., Eq. (76), Eq. (78) and Eq. (84), w.h.p., we have

$$\|\hat{R}_3^{(k)} - R_3^{(k)}\|_{op} \leq \delta_3 \|\boldsymbol{w}_k^*\|^3$$

Now we can complete the proof by combing all the $K$ components, w.p. $1 - O(Kd^{-2})$

$$\|\hat{R}_3 - R_3\|_{op} \leq \sum_{k \in [K]} p_k \|\hat{R}_3^{(k)} - R_3^{(k)}\|_{op} \leq \delta_3 \sum_{k \in [K]} p_k \|\boldsymbol{w}_k^*\|^3 \tag{85}$$

$\square$

### B.4.5  Proof of Lemma 12

*Proof.* Most part of the proof follows the proof of Lemma 4 in [7]. Let $\hat{W}^T R_2 \hat{W} = U \Lambda U^T$. Define $W := \hat{W} U \Lambda^{-1/2} U^T$, then $W$ is the whitening matrix of $R_2$, i.e., $W^T R_2 W = I$. Define the whitened tensor $T = R_3(W, W, W)$, i.e.,

$$
\begin{aligned}
T &:= \sum_{k=1}^{K} p_k W^T \boldsymbol{u}_k \otimes W^T \boldsymbol{u}_k \otimes W^T \boldsymbol{u}_k \\
&= \sum_{k=1}^{K} p_k^{-1/2} (p_k^{1/2} W^T \boldsymbol{u}_k) \otimes (p_k^{1/2} W^T \boldsymbol{u}_k) \otimes (p_k^{1/2} W^T \boldsymbol{u}_k) \\
&= \sum_{k=1}^{K} p_k^{-1/2} \boldsymbol{v}_k \otimes \boldsymbol{v}_k \otimes \boldsymbol{v}_k,
\end{aligned} \tag{86}
$$

where $\{\boldsymbol{v}_k := p_k^{1/2} W^T \boldsymbol{u}_k\}_{k=1}^{K}$ are orthogonal basis because $\sum_{k=1}^{K} \boldsymbol{v}_k \boldsymbol{v}_k^T = W^T R_2 W = I_K$. In practice, we have $\hat{T} := \hat{M}_3(\hat{W}, \hat{W}, \hat{W})$, an estimation of $T$. Define $\epsilon_T := \|\hat{T} - T\|_{op}$. Similar to the proof of Lemma 4 in [7], we have

$$
\begin{aligned}
\epsilon_T =& \|R_3(W, W, W) - \hat{R}_3(\hat{W}, \hat{W}, \hat{W})\|_{op} \\
\leq& \|R_3(W, W, W) - R_3(W, W, \hat{W})\|_{op} + \|R_3(W, W, \hat{W}) - R_3(W, \hat{W}, \hat{W})\|_{op} \\
&+ \|R_3(W, \hat{W}, \hat{W}) - R_3(\hat{W}, \hat{W}, \hat{W})\|_{op} + \|R_3(\hat{W}, \hat{W}, \hat{W}) - \hat{R}_3(\hat{W}, \hat{W}, \hat{W})\|_{op} \\
=& \|R_3(W, W, W - \hat{W})\|_{op} + \|R_3(W, W - \hat{W}, \hat{W})\|_{op} \\
&+ \|R_3(W - \hat{W}, \hat{W}, \hat{W})\|_{op} + \|R_3(\hat{W}, \hat{W}, \hat{W}) - \hat{R}_3(\hat{W}, \hat{W}, \hat{W})\|_{op} \\
\leq& \|R_3\|_{op}(\|W\|^2 + \|W\|\|\hat{W}\| + \|\hat{W}\|^2)\epsilon_W + \|\hat{W}\|^3 \epsilon_3
\end{aligned} \tag{87}
$$

where $\epsilon_W = \|\hat{W} - W\|$.

If $\epsilon_2 \leq \sigma_K/3$, we have $|\sigma_K(\hat{R}_2) - \sigma_K| \leq \epsilon_2 \leq \sigma_K/3$. Then $\frac{2}{3}\sigma_K \leq \sigma_K(\hat{R}_2) \leq \frac{4}{3}\sigma_K$ and $\|\hat{W}\| \leq \sqrt{2}\sigma_K^{-1/2}$.

$$\epsilon_W = \|\hat{W} - W\| = \|\hat{W}(I - U\Lambda^{-1/2}U^T)\| \leq \|\hat{W}\|\|I - \Lambda^{-1/2}\| \tag{88}$$

Since we have $\|I - \Lambda\| = \|\hat{W}^T R_2 \hat{W} - \hat{W}^T \hat{R}_2 \hat{W}\| \leq \|\hat{W}\|^2 \epsilon_2 = 2\sigma_K^{-1}\epsilon_2$. Thus

$$\|I - \Lambda^{-1/2}\| \leq \max\{|1 - (1 + 2\epsilon_2/\sigma_K)^{-1/2}|, |1 - (1 - 2\epsilon_2/\sigma_K)^{-1/2}|\} \leq \epsilon_2/\sigma_K$$

Therefore,

$$\epsilon_W \leq \sqrt{2}\epsilon_2 \sigma_K^{-3/2} \tag{89}$$

Now we have

$$\epsilon_T \leq 8\|R_3\|_{op}\sigma_K^{-5/2}\epsilon_2 + 2\sqrt{2}\sigma_K^{-3/2}\epsilon_3 \tag{90}$$

Thus we can apply Theorem 5.1 [2] to show the guarantees of the robust tensor power method to recover $\{\boldsymbol{v}_k\}_{k=1}^{K}$ and $\{p_k\}_{k=1}^{K}$. It can be stated as below, for some universal constant $c_T$ and a small

value $\eta$ (the computational complexity is related to $\eta$ by $O(\log(1/\eta))$), if $\epsilon_T \leq c_T \frac{1}{K\sqrt{p_{max}}}$, w.p. $1 - \eta$ the returned eigenvectors $\{\hat{\boldsymbol{v}}_k\}_{k=1}^K$ and eigenvalues $\{\hat{a}_k\}_{k=1}^K$ satisfy

$$\|\hat{\boldsymbol{v}}_k - \boldsymbol{v}_k\| \leq 8\epsilon_T \sqrt{p_k} \leq 8\epsilon_T \sqrt{p_{max}}, \ |\hat{a}_k - \frac{1}{\sqrt{p_k}}| \leq 5\epsilon_T \tag{91}$$

Let $a_k = \frac{1}{\sqrt{p_k}}$. Now we show

$$
\begin{aligned}
\|(\hat{W}^T)^\dagger(\hat{a}_k\hat{\boldsymbol{v}}_k) - \boldsymbol{u}_k\| &= \|(\hat{W}^T)^\dagger(\hat{a}_k\hat{\boldsymbol{v}}_k) - W^\dagger a_k\boldsymbol{v}_k\| \\
&\leq \|(\hat{W}^T)^\dagger(\hat{a}_k\hat{\boldsymbol{v}}_k) - (\hat{W}^T)^\dagger(a_k\boldsymbol{v}_k)\| + \|(\hat{W}^T)^\dagger(a_k\boldsymbol{v}_k) - (W^T)^\dagger a_k\boldsymbol{v}_k\| \\
&\leq \|(\hat{W}^T)^\dagger\|(\|\hat{a}_k\hat{\boldsymbol{v}}_k - \hat{a}_k\boldsymbol{v}_k\| + \|\hat{a}_k\boldsymbol{v}_k - a_k\boldsymbol{v}_k\|) + \|(\hat{W}^T)^\dagger - (W^T)^\dagger\|\|a_k\boldsymbol{v}_k\| \\
&\leq \|(\hat{W}^T)^\dagger\|(\hat{a}_k 8\epsilon_T/a_k + 5\epsilon_T) + \|(\hat{W}^T)^\dagger - (W^T)^\dagger\|a_k
\end{aligned}
\tag{92}
$$

If $\epsilon_T \leq \frac{1}{10\sqrt{p_{max}}}$, we have $\hat{a}_k/a_k \leq 3/2$. If $\epsilon_2 \leq \sigma_K/3$,

$$\|(\hat{W}^T)^\dagger\| = \|\hat{\Lambda}_2\|^{1/2} \leq \sqrt{2}\|R_2\|^{1/2} \tag{93}$$

and

$$
\begin{aligned}
\|(\hat{W}^T)^\dagger - (W^T)^\dagger\| &= \|(\hat{W}^T)^\dagger(I - U\Lambda^{1/2}U^T)\| \\
&= \|(\hat{W}^T)^\dagger\|\|I - \Lambda^{1/2}\| \\
&\leq 2\sqrt{2}\|R_2\|^{1/2}\epsilon_2/\sigma_K
\end{aligned}
\tag{94}
$$

$$
\begin{aligned}
\|(\hat{W}^T)^\dagger(\hat{a}_k\hat{\boldsymbol{v}}_k) - \boldsymbol{u}_k\| &\leq \|R_2\|^{1/2}(25\epsilon_T + 3\epsilon_2/\sigma_K) \\
&\leq (3\|R_2\|^{1/2}\sigma_K^{-1} + 200\|R_2\|^{1/2}\|R_3\|_{op}\sigma_K^{-5/2})\epsilon_2 + (75\|R_2\|^{1/2}\sigma_K^{-3/2})\epsilon_3
\end{aligned}
\tag{95}
$$

$\square$

## C  Proofs of Subspace Clustering

### C.1  Some Properties of the Distance between Subspaces

According to [16], $D(U,V) = \sqrt{r - \|U^T V\|_F^2} = \sqrt{\text{tr}(I_r - U^T V V^T U)} = \|U_\perp^T V\|_F = \|V_\perp^T U\|_F$. We briefly give the proof.

$$\|UU^T - VV^T\|_F^2 = \|(I - VV^T)UU^T - VV^T(I - UU^T)\|_F^2$$

Since $(I - VV^T)UU^T(VV^T(I - UU^T))^T = 0$ and $(VV^T(I - UU^T))^T(I - VV^T)UU^T = 0$, we have

$$
\begin{aligned}
&\|(I - VV^T)UU^T - VV^T(I - UU^T)\|_F^2 \\
&= \|(I - VV^T)UU^T\|_F^2 + \|VV^T(I - UU^T)\|_F^2 \\
&= 2\,\text{tr}(VV^T - V^T UU^T V) \\
&= 2(r - \|V^T U\|_F^2)
\end{aligned}
\tag{96}
$$

By the property of Frobineous norm we see $D(\cdot,\cdot)$ is a metric, so we can use triangular inequality. We will also use the following inequality, which is due to the dual property of matrices, ref. Lemma 3.2 in [5]. Let $A, B$ be two matrices.

$$
\begin{aligned}
&\|AB\|_F \\
&= \langle AB, AB\rangle^{1/2} \\
&= \langle BB^T, A^T A\rangle^{1/2} \\
&\leq \|BB^T\|^{1/2}\|A^T A\|_*^{1/2} \\
&= \|B\|\|A\|_F
\end{aligned}
\tag{97}
$$

Similarly, we also have $\|AB\|_F \leq \|A\|\|B\|_F$.

## C.2 Proof of Theorem 5

*Proof.* For simplicity, we use $U_j$ to denote $U_j^t$ for $j \in [K]$. Consider fixing $\{U_k\}_{k \neq j}$ and updating $U_j$.

$$
\begin{aligned}
\bar{U}_j &= \sum_{i=1}^{N} \left( \Pi_{k \neq j} \langle I_d - U_k U_k^T, \boldsymbol{z}_i \boldsymbol{z}_i^T \rangle \right) \boldsymbol{z}_i \boldsymbol{z}_i^T U_j \\
&= \sum_{q=1}^{K} \sum_{i \in \Omega_q^{(t)}} \left( \Pi_{k \neq j} \langle I_d - U_k U_k^T, U_q^* \boldsymbol{s}_i \boldsymbol{s}_i^T U_q^{*T} \rangle \right) U_q^* \boldsymbol{s}_i \boldsymbol{s}_i^T U_q^{*T} U_j \\
&= \sum_{q=1}^{K} U_q^* \sum_{i \in \Omega_q^{(t)}} \left( \Pi_{k \neq j} \langle I_r - U_q^{*T} U_k U_k^T U_q^*, \boldsymbol{s}_i \boldsymbol{s}_i^T \rangle \right) \boldsymbol{s}_i \boldsymbol{s}_i^T U_q^{*T} U_j
\end{aligned}
\tag{98}
$$

where $\Omega_q^{(t)}$ is the set of data points belongs to $q$-th subspace in $t$-th iteration.

Define

$$
B_{jq} := \mathbb{E}\left[ \Pi_{k \neq j} (\boldsymbol{s}^T (I_r - U_q^{*T} U_k U_k^T U_q^*) \boldsymbol{s}) \boldsymbol{s} \boldsymbol{s}^T \right]
\tag{99}
$$

$$
\hat{B}_{jq} := \frac{1}{|\Omega_q^{(t)}|} \sum_{i = \Omega_q^{(t)}} \Pi_{k \neq j} (\boldsymbol{s}_i^T (I_r - U_q^{*T} U_k U_k^T U_q^*) \boldsymbol{s}_i) \boldsymbol{s}_i \boldsymbol{s}_i^T
\tag{100}
$$

According to Lemma 3, we have

$$
\Pi_{k \neq j} \operatorname{tr} (I_r - U_q^{*T} U_k U_k^T U_q^*) I \preceq B_{jq} \preceq C_{K-1} \Pi_{k \neq j} \operatorname{tr} (I_r - U_q^{*T} U_k U_k^T U_q^*) I
\tag{101}
$$

Note that $\operatorname{tr}(I_r - U_q^{*T} U_k U_k^T U_q^*) = D(U_q^*, U_k)^2$. We have

$$
\Pi_{k \neq j} D(U_q^*, U_k)^2 I \preceq B_{jq} \preceq C_{K-1} \Pi_{k \neq j} D(U_q^*, U_k)^2 I
\tag{102}
$$

If the conditions about $n$ and $r$ in Theorem 5 are satisfied, because of Lemma 4 with $\delta = \frac{1}{2C_{K-1}}$, we have, w.p. $1 - O(Kr^{-2})$,

$$
\|B_{jq} - \hat{B}_{jq}\| \leq \frac{1}{2C_{K-1}} \|B_{jq}\| \leq \frac{1}{2} \Pi_{k \neq j} \operatorname{tr} (I_r - U_q^{*T} U_k U_k^T U_q^*) \leq \frac{1}{2} \sigma_{min}(B_{jq})
$$

Therefore,

$$
\frac{1}{2} \Pi_{k \neq j} \operatorname{tr} (I_r - U_q^{*T} U_k U_k^T U_q^*) I \preceq \hat{B}_{jq} \preceq (C_{K-1} + 1) \Pi_{k \neq j} \operatorname{tr} (I_r - U_q^{*T} U_k U_k^T U_q^*) I
\tag{103}
$$

Given the condition, $D(U_k^*, U_k) \leq c_s \min_{q \neq j} \{D(U_q^*, U_j^*)\}$, we have for $q \neq k$

$$
D(U_q^*, U_k)^2 \leq (D(U_q^*, U_k^*) + D(U_k^*, U_k))^2 \leq (1 + c_s)^2 D(U_q^*, U_k^*)^2
$$

Similarly,

$$
D(U_q^*, U_k)^2 \geq (1 - c_s)^2 D(U_q^*, U_k^*)^2
$$

Therefore, for $j \neq q$

$$
\|\hat{B}_{jq}\| \leq (C_{K-1} + 1) \left( \Pi_{k:k \neq j, k \neq q} (1 + c_s)^2 D(U_q^*, U_k^*)^2 \right) D(U_q^*, U_q)^2
\tag{104}
$$

For $j = q$

$$
\sigma_{min}(\hat{B}_{jj}) \geq \frac{1}{2} \Pi_{k:k \neq j} (1 - c_s)^2 D(U_j^*, U_k^*)^2
\tag{105}
$$

Shown in Eq. (98), $\bar{U}_j = \sum_{q=1}^{K} U_q^* p_q \hat{B}_{jq} U_q^{*T} U_j$. Let $[U_j^+, \bar{R}_j] := \text{QR}(\bar{U}_j)$

$$D(U_j^+, U_j^*)$$
$$= \|U_{j\perp}^{*T} \bar{U}_j \bar{R}_j^{-1}\|_F$$
$$\leq \|U_{j\perp}^{*T} \bar{U}_j\|_F \|\bar{R}_j^{-1}\|$$
$$= \|U_{j\perp}^{*T} \sum_{q=1}^{K} U_q^* p_q \hat{B}_{jq} U_q^{*T} U_j\|_F \|\bar{R}_j^{-1}\|$$
$$\leq (\sum_{q \neq j} p_q \|U_{j\perp}^{*T} U_q^*\|_F \|\hat{B}_{jq}\| \|U_q^{*T} U_j\|) \|\bar{R}_j^{-1}\|$$
$$\leq (\sum_{q \neq j} p_q \|U_{j\perp}^{*T} U_q^*\|_F \|\hat{B}_{jq}\|) \|\bar{R}_j^{-1}\|$$
$$\leq (C_{K-1} + 1)(\sum_{q \neq j} p_q D(U_j^*, U_q^*)\Big((1+c_s)^{2(K-2)} \Pi_{k:k \neq j, k \neq q} D(U_q^*, U_k^*)^2\Big) D(U_q^*, U_q)^2) \|\bar{R}_j^{-1}\|$$
$$\leq (C_{K-1} + 1)(1+c_s)^{2(K-2)} \left( \sum_{q \neq j} p_q D(U_j^*, U_q^*)\big(\Pi_{k:k \neq j, k \neq q} D(U_q^*, U_k^*)^2\big) D(U_q^*, U_q)^2 \right) \|\bar{R}_j^{-1}\|$$
$$\leq (C_{K-1} + 1)(1+c_s)^{2(K-2)} p_{max}(K-1) D_{max}^{2K-3} D(U_q^*, U_q)^2 \|\bar{R}_j^{-1}\|$$
$$\tag{106}$$

Now we show,
$$\|\bar{R}_j^{-1}\| \leq \sigma_{min}^{-1}(\bar{R}_j) = \sigma_{min}^{-1}(\bar{U}_j) \leq (p_j \sigma_{min}(\hat{B}_{jj}))^{-1}$$
$$\leq \big((p_j/2) \Pi_{k:k \neq j} (1-c_s)^2 D(U_j^*, U_k^*)^2\big)^{-1}$$
$$\leq \frac{2}{p_{min}(1-c_s)^{2K-2} D_{min}^{2K-2}}$$
$$\tag{107}$$

Combing Eq.(106), Eq. (107) and the condition on $c_s$,

$$D(U_j^+, U_j^*) \leq 2(C_{K-1} + 1)(K-1)(1+c_s)^{2(K-2)}(1-c_s)^{-2(K-1)} \frac{p_{max} D_{max}^{2K-3}}{p_{min} D_{min}^{2K-2}} D(U_q^*, U_q)^2$$
$$\leq \frac{1}{2c_s D_{min}} \Delta_t^2$$
$$\tag{108}$$

Using the initialization condition, we can easily obtain $\frac{\Delta_t}{2c_s D_{min}} \leq \frac{1}{2}$ by induction. Also, the condition, $D(U_j^+, U_j^*) \leq c_s D_{min}$, still holds after each update. So we have super-linear convergence rate. $\qquad\square$

## D    More Experimental Results

In Fig. 2, we show that, to achieve an initial error $\epsilon^{(0)} = c$ for some constant $c < 1$, our tensor method only requires $N$ to be proportional to $d$. Note that the naive initialization methods, random initialization (using normal distribution) or all-zero initialization, will lead to $\epsilon^{(0)} \approx 1.4$ and $\epsilon^{(0)} = 1$ respectively.

In Fig. 3 we compare our methods with EM in terms of iterations. In Fig. 4 we compare EM and our methods for larger $K$, $K = 6$. Note that the per-iteration cost of MLR will be $K$ times more than the per-iteration cost of EM. So when $K$ is larger, MLR will be slower than EM.

In Fig. 5, we show the sample complexities for different methods. Our methods (MLR) have a better sample complexity than EM. And the tensor initialization outperforms random initialization significantly.

Fig. 6 shows whatever the ambient dimension $d$ is, the clusters will be exactly recovered when $N$ is proportional to $r$ by a constant factor.

Figure 2: Initialization error for tensor method.

(a) $d = 100$, $N = 6k$, $K = 3$

(b) $d = 1k$, $N = 60k$, $K = 3$

Figure 3: Comparison with EM in terms of iterations.

Table 2: Corresponding CE's for the results in Table 1

| $N/K$ | SSC | SSC-OMP | LRR | TSC | NSN+spectral | NSN+GSR | **PSC** |
|-------|-----|---------|------|------|--------------|---------|---------|
| 200 | 0 | 0.0190 | 0.0010 | 0.0650 | 0 | 0 | 0 |
| 400 | 0 | 0.0090 | 0.0015 | 0.0190 | 0 | 0 | 0 |
| 600 | 0 | 0.0027 | 0 | 0.0120 | 0 | 0 | 0 |
| 800 | 0 | 0.0027 | 0 | 0.0030 | 0 | 0 | 0 |
| 1000 | 0 | 0.0014 | 0 | 0.0022 | 0 | 0 | 0 |

(c) $d = 60$, $N = 36k$, $K = 6$

(d) $d = 300$, $N = 180k$, $K = 6$

Figure 4: Comparison with EM for larger $K$, $K = 6$

(a) EM with random initialization

(b) EM with tensor initialization

(a) MLR with random initialization

(b) MLR with tensor initialization

Figure 5: Sample complexities for different methods

Figure 6: Subspace Clustering error for different $N$, $d$ and $r$.