[Reviews · NeurIPS 2016]

Reviewer 1

Summary

This paper studies the mixed linear regression problem where the goal is to recover multiple linear models given a dataset of samples and unlabeled responses. The authors propose a smooth non-convex objective function and show under some statistical assumptions that it is locally strongly convex in the neighborhood of the optimal solution. They use a tensor method to provide initialization which is in the locally strongly convex region after which standard convex optimization algorithms can be used such as (accelerated) gradient descent. They provide the first global recovery guarantees in the noiseless setting for problems with more than 2 models (components) and show their method has almost optimal computational complexity O(Nd) where N is the number of samples and d is the dimensionality and almost optimal sample complexity O(d(K\log{d})^K) where K is the number of models. Moreover, they show their non-convex formulation can be used for subspace clustering where their method converges to the global optima in time linear in the number of points when initialized a small constant away from the true subspaces. The authors provide experimental results on synthetic data where they compare their method with EM using both tensor initialization and random initialization. The experiments support the theory which shows their method converges faster than EM which, in some cases, may not converge at all. Moreover, the experimental results support the theory which shows the sample complexity grows almost linearly with the dimensionality.

Qualitative Assessment

The work is an advance in the context of mixed linear regression, and the paper is well written. Some comments - 1. Is the exponential dependence on K in the sample complexity avoidable? This seems like a downside of the proposed method. 2. Can your analysis be adapted to show similar recovery guarantees in noisy settings? The model in (3) assumes y = x^T w, how does the analysis change if say thin-tailed or bounded noise is added. 3. Will your analysis only hold when samples are drawn from a Gaussian distribution with identity covariance matrix? What about general sub-Gaussian distributions? Note that this determines the scope of applicability of the method. 4. Why must you assume that \Delta_min / \Delta_max is independent of d? Is this a reasonable assumption? Further, is Assumption 1 reasonable? 5. It seems as if your results, e.g., Theorem 1, fail with constant probability. Can you provide high-probability results? In particular, can the dependency on N (possibly exp(-N) or equivalent for N > N_0) be made explicit. 6. In the last column in Table 1, why does the time decrease when N/K goes from 200 to 400 then increase for the rest? Are these results average times? What are the standard deviations? 7. Typo in (1) - the sum should run to N number of samples not n.

Confidence in this Review

2-Confident (read it all; understood it all reasonably well)


Reviewer 2

Summary

This paper proposes a non convex method to optimize a surrogate of the objective function of the mixed linear regression problem when the number of components is fixed and known. Although the surrogate objective is non-convex, it is proved locally strongly convex, which paves the way for efficient optimization and theoretical analysis. The optimization algorithm consists in two important blocks: first, an initialization step achieved via an tensor method inspired from the existing literature. Second, a simple gradient descent. Theoretical guarantees for the global convergence of the algorithm are given, which require a not too large number of components compared to the sample size. The precision, however, is independent of the sample size. The global computational complexity is shown to be nearly optimal in O(dnlog(d/eps)) where d is the problem dimension, n the sample size and eps the required precision. Numerical comparison is made with the more classical EM strategy on synthetic data. This shows that all methods perform similarly in terms of both statistical performance and computational performance when they are initialized with the tensor approach. Finally, an interesting extension of the method to subspace clustering is made, showing very competitive running times compared to the state-of-the-art approaches.

Qualitative Assessment

pros: well written, very clear presentation with a good introduction and appropriate bibliography. Good balance between theoretical justifications, algorithms and numerical experiments. An effort is made for broadening the method, beyond MLR, by considering the Subspace Clustering problem. Sketches of proofs are provided to avoid excessive technicality, postponed to supplementary material. cons: no real-world data sets is analyzed, neither with the MLR nor the SC. Their is no clear practical advantage over the EM algorithm when the correct initialization is chosen. On the theoretical side, however, the authors' proposal has interesting justification. Moreover, the Subspace clustering appears to be a even more interesting application of the method than MLR.

Confidence in this Review

2-Confident (read it all; understood it all reasonably well)


Reviewer 3

Summary

This paper proposes a new approach to mixed linear regression. The idea is that we observe (yi,xi) pairs, and there is some partitioning of these pairs so that for each partition set there is a good linear regressor. The authors propose estimating this partition and associated regressors by minimizing a novel, non-convex objective function. They also develop an efficient tensor-based initialization scheme which, combined with gradient descent on the proposed objective, leads to the globally optimal solution. They also establish a connnection with subspace clustering and propose a novel objective function for solving that problem.

Qualitative Assessment

I really liked this paper. They consider a challenging problem, propose a novel and not immediately obvious solution, and compelling theory to support their approach. There are several grammatical errors throughout, but otherwise the paper is clear and well written, and does an excellent job of putting the paper in the proper context. The experimental results focus on time complexity. It would have been nice to see something about sample complexity, which is addressed in the paper, as well. Figure 1(a) covers sample complexity for the proposed approach, but there are no comparisons with the plethora of other work in this space. In addition, the proposed objective function seems very reasonable in noiseless or low-noise settings, but the stability of the proposed method is very unclear to me. In higher noise settings, I wonder if the proposed tensor initialization followed by the EM algorithm would be less sensitive to noise.

Confidence in this Review

2-Confident (read it all; understood it all reasonably well)


Reviewer 4

Summary

This paper proposes a new objective function to solve mixed linear regression problem in the multiple component and noiseless case. The author(s) shows theoretically that using a tensor method as initialization, their estimator is guaranteed to have global convergence. And their method has near-optimal computational complexity and computational complexity when component number K is fixed. The experiment shows that the solution error converge faster using the proposed method comparing one EM method, and without tensor method as initialization method, EM method may not converge. The author(s) also extend their method to subspace clustering. And their method is faster than other subspace clustering methods, empirically.

Qualitative Assessment

This paper proposes a new objective function to solve mixed linear regression problem, but fails to explain many important issues: (1) What is the intuition of the introduction and advantage of the objective function? The answer between line 39 and line 40 is not good. Because if it is modeled as finite mixture model as in many references, "objective value is zero when {w_k}_{k=1,2,...,K} is the global optima and y’s do not contain any noise" is also true. The following is a example. Städler, Nicolas, Peter Bühlmann, and Sara Van De Geer. "ℓ 1-penalization for mixture regression models." Test 19.2 (2010): 209-256. (2) What is the relationship between the model in Eq.(3) and the objective function in Eq.(1)? It seems there is no probabilistic interpretation for the objective function in Eq.(1). (3) Between line 40 and 41, the author(s) says "the objective function is smooth and hence less prone to getting stuck in arbitrary saddle points or oscillating between two points". Why is that? As we know, many deep learning objective functions are smooth and still very easy to get stuck in saddle points or cause oscillation. (4) The explanations and presentation of EM methods are very poor. Firstly, this paper talks about EM methods all the time without giving a specific objective function and algorithm to clarify what it is talking about. Secondly, to compare the objective function proposed by this paper, the author(s) should use another objective function to make a valid comparison. But EM is basically a algorithm, not a objective function. Maybe the author should compare the objective function of finite mixture regression which is summarized in the following reference. Khalili, Abbas, and Jiahua Chen. "Variable selection in finite mixture of regression models." Journal of the american Statistical association (2012). But the objective function of finite mixture regression does not "makes a 'sharp' selection of mixture component", as stated in line 42 in this paper. So I finally find out that the mentioned EM method is more likely the method used in the following reference, which is the 28th reference of this paper. Yi, Xinyang, Constantine Caramanis, and Sujay Sanghavi. "Alternating Minimization for Mixed Linear Regression." ICML. 2014. However, the so called EM method used in the above reference uses a "hard" choice method in the E-step, which makes this method not a EM algorithm, rather a kmeans-like algorithm. And this maybe the reason that the author(s) claims that EM method "makes a 'sharp' selection of mixture component". Lastly, in Figure 1(d), how many times did the experiment run? Without tensor method as initialization method, does the so-called EM method always not converge? (5) The author(s) should explain when Assumption 1 will hold.

Confidence in this Review

2-Confident (read it all; understood it all reasonably well)


Reviewer 5

Summary

This paper studies the two related problems of mixed linear regression (MLR) and subspace clustering. The authors propose non-convex optimization models for the problems. For MLR, it is shown that the model is locally strongly convex, and an initialization algorithm is proposed so that the overall method has global convergence guarantees. For subspace clustering, a power method is proposed for solving the nonconvex optimization problem, which is shown to converge to the global optima provided that the initialization is within a small distance to the true solutions. Another benefit of the proposed subspace clustering method is that it is of linear complexity in the number of data points, and experiments validates its efficiency.

Qualitative Assessment

The subspace clustering part of the work presents a linear-time algorithm, and a significant contribution of this work is that the algorithm is proved to converge to the ground truth solutions superlinearly provided that the initialization is good enough. This result may be of great interest to the subspace clustering community. However, a critical issue with the discussion of this method is that the authors did not mention the very closely related method of K-subspaces, thus the current discussion of the method is incomplete and problematic. The K-subspace method is a linear time subspace clustering method which alternates between estimating the subspaces and estimating the data membership (akin to K-means), see, e.g. the recent textbook Generalized PCA by Rene Vidal, 2016, or the paper [a] R. Vidal, A tutorial on subspace clustering, 2011. Although it has linear complexity, maybe the method has mostly been abandoned recently due to the fact that it is hard to initialize the subspaces, and that it suffers from the model selection issue (e.g. how to choose the dimension of the subspaces). The objective function (2) is very similar to that of the K-subspaces: it also has linear complexity, and it is also likely to suffer from the same issues. Thus, the authors should at least mention K-subspaces as a closely related method, and comment on the contribution of this work on top of that, and if possible, comment on how the drawbacks of K-subspaces are addressed by the proposed method. Although I tend to believe that the proposed method has the same drawback and thus is unlikely to give better performance for real applications, the result of theorem 5 is interesting and is worth to be published, provided that the related K-subspace method is discussed properly. Some detailed comments: - Line 14-15 "a significant step towards solving the open problem of subspace clustering in linear time": It is inappropriate to call it an open problem because of the K-subspaces method mentioned above; besides, there are also other linear time methods [b] A. Adler, M. Elad, Linear-Time subspace clustering via bipartite graph modeling, TNNLS 2013 [c] J. Shen, P. Li, H. Xu, Online low-rank subspace clustering by basis dictionary pursuit, ICML 2016. - Line 60 “solving MLR using SC is intractable because the dimension of each subspace is only one less than the ambient dimension”: this comment may be true for most of the recent spectral clustering based methods as mentioned by the authors. However, there are other methods (see e.g. [a]) that are able to deal with high rank cases, e.g., [d] R. Vidal et. al., Generalized Principal Component Analysis, TPAMI 2005 - Line 268 "no assumptions are needed on the affinity between subspaces": I don't see why this is true since the affinity between subspaces affects the conditions in theorem 5. Specifically, according to eq. (9), is it "easier" to initialize the algorithm if subspaces are more separated? - Line 276: it will look better to put space between alg. 2 and 3. - Section 5.2: The "LLR" should be LRR. For SSC-OMP, another work is [e] C. You et. al., Scalable sparse subspace clustering by orthogonal matching pursuit, ArXiv'15/CVPR'16. - Table 1: at least the K-subspace method should be added as a comparison. Also, the reported result for SSC-OMP is slower than what is commonly observed in [23] and [e]. An efficient implementation is at http://vision.jhu.edu/code/. It would also be interesting to see results on larger data, e.g., with N = 100,000 to 1,000,000 as in [b,e], which would also strengthen the claim of the efficiency of the algorithm. To summarize, I have some low rating for novelty, potential impact and proper references primarily due to the fact that the prior work for subspace clustering is not properly commented and there are several claims that may be problematic. I would be subject to change if the author address the issues properly.

Confidence in this Review

2-Confident (read it all; understood it all reasonably well)


Reviewer 6

Summary

The paper under review studies the mixed linear regression problem, where the goal is to recover multiple underlying linear models from their unlabeled linear measurements. The paper proposes a non-convex objective function, which, as shown in the paper, is close to being locally strongly convex in the neighborhood of the ground truth, and use a tensor method for initialization so that the initial modes actually fall into this strongly convex regime. Specifically, the paper assumes pairs (y_i,x_i) to be generated as y_i = < w_{z_i},x_i > , where z_i is an multinomial distributed index, w_k is the latent parameter of the k-th mixture component, and x_i is Gaussian. Theorem 1 guarantees that provided the parameters w_k are independent of the samples and lie in the neighborhood of the solution, the Hessian is positive semi-definite (PSD) at the w_k with high probability. This does not imply, however, that the function is strongly convex around the true solution, as the statement is only for the Hessian at a set of given w_k (i.e., at one point, a set of w_k corresponds to one point in the parameter space). Theorem 2 proves a similar statement than Theorem 2, this time, however, the w_k are not assumed to be independent of the samples. The cost for this stronger statement is that the w_k need to be significantly closer to the w_k^ast. Again, the statement is for a given set of w_k, and does not hold for all w_k in a ball with high probability. From those statements, the paper concludes that certain convex algorithms such as gradient descent converge to the true solution. For this conclusion to be true, more has to be shown; I don't see how from point wise strongly convexity, convergence follows (why don't we need strong convexity for all points in a ball w.h.p.?). In section 3.3, the paper proposes a tensor method to initialize the parameters. This initialization is interesting in that it provides an initialization that falls within the radius of point wise strong convexity of Theorem 1. Additionally, it has complexity O(KdN + K^3 N poly(K) ). Thus, the complexity is small if K is small; however, for moderate to large K, the complexity is large. In Section 4, the paper consider extensions to the subspace clustering problem. Finally, the paper provides simulation results where the data points y_i are corrupted by additive noise.

Qualitative Assessment

Overall I think this paper presents an interesting approach, well suited for NIPS. Below are some comments and and also concerns. 1. My strongest concern is on whether Theorem 1 and 2 indeed guarantee convergence of algorithms like gradient descent. Specifically, Theorem 1 and 2 show that with high probability, for a given point {w_k}_k sufficiently close to {w_k^ast}_k, the Hessian is positive semi definite (PSD)---as the paper puts it ``strongly convex at a point''. However the definition of a parameter converging algorithm (Def. 1) and the standard convergence proofs of gradient descent require the function to be strongly convex in a region, not only at a point. Note that Theorem 1 and 2 do not guarantee that the Hessian is PSD at all points simultaneously. Maybe Theorem 1 and 2 can be used together with an epsilon-net argument and appropriate union bounds to show that the function is strongly convex in a certain region around the optimum. However this is not done in the current version of the paper. It is therefore not clear to me whether Corollary 1 and the conclusion in Section 3.1 hold. Could the paper please address this point by providing a formal proof of Corollary 1 (i.e., of the statement of Theorem 1 implying convergence of say gradient descent)? 2. The proofs are not very formal. Specifically, Appendix A states several lemmas guaranteeing that certain events hold with high probability, and, e.g., the proof of Theorem 1 simply assumes some of those events throughout the proof. However, the proof does not explain why the assumptions of the lemmata hold, and does not put together the events via say the union bound. Additionally, in the derivations the paper does not explain where exactly the assumptions come in. This makes it difficult and time consuming to verify the correctness of the proofs. 3. Overall the paper is well written; the proofs however miss proper punctuation throughout. 4. Theorem 3 makes some abstract assumptions on the original w_k^ast (Assumption 1). Could the paper give some intuition for which constellations of w_k those assumptions will be satisfied, and for which they fail? E.g., by discussing examples. 5. Theorem 5 on subspace clustering assumes that the subspaces are already initialized in a way so they are very, very close to the true solution (Eq. (9)) (exponentially small in K!). That seems to be a very stringent condition, as it is not clear how one finds such an initialization. 6. Subspace clustering: It is very interesting that no conditions on the affinity of the subspaces are needed. Is the intuition for this result, that the paper does not consider actual clustering, but subspace recovery? E.g., if two subspaces are exactly the same, this would not be an issue for subspace recovery, as considered in the paper, but would make the clustering problem ill posed as a given point could be assigned to one or the other subspace. 7. The numerical results show that the algorithm does very well in terms of running time, which is great. However, the results are for K=3. For larger K, one would expect the results to be different, specifically, the other algorithms would, for large enough K, perform better in terms of computation time than the proposed algorithm PSC. It would be interesting to show simulation for larger values of K, to see for which K PSC is faster, and for which it is not. This would also illustrate that PSC is not uniformly better.

Confidence in this Review

2-Confident (read it all; understood it all reasonably well)